# PerceptionRubrics: Calibrating Multimodal Evaluation to Human Perception

**Yana Wei** [*1]  **Hongbo Peng** [*2]  **Yanlin Lai** [*3]  **Liang Zhao** [2]  **Kangheng Lin** [2]  **En Yu** [2]  **Keyu Lv** [2]  **Han Zhou** [2]
**Yin Tang** [4]  **Haodong Li** [2]  **Mitt Huang** [2]  **Hangyu Guo** [2]  **Jianjian Sun** [2]  **Zheng Ge** [2]  **Xiangyu Zhang** [2]
**Daxin Jiang** [2]  **Vishal Patel** [1]

## Abstract

We introduce PERCEPTIONRUBRICS, a rubric-based evaluation framework that addresses the dissonance between benchmark saturation and real-world brittleness. Shifting evaluation from holistic semantic matching to rigorous atomic auditing, PERCEPTIONRUBRICS pairs 1,038 information-dense images with over 12,000 instance-specific rubrics. These criteria are derived from golden captions that constructed via a novel Circular Peer-Review consensus pipeline and then distilled into a dual-stream system of *Must-Right* (essential facts) and *Easy-Wrong* (fine-grained details) rubrics. Crucially, PERCEPTIONRUBRICS implements a Gated Scoring mechanism: unlike linear averages, failure on mandatory visual facts triggers sharp binary penalties. Extensive evaluation yields critical insights: (1) **The Reliability Gap**: models often verify fragmented elements correctly yet fail strict conjunctive constraints, exposing brittleness in dense domains; (2) **Open-Closed Stratification**: contrary to reasoning trends, we reveal a persistent 5% perception deficit between open-source and proprietary frontiers; and (3) **Human-Aligned Rigor**: our gated metrics substantially out-align conventional benchmarks, validating that strict perceptual fidelity is the prerequisite for reliable generation.

## 1. Introduction

Despite the rapid evolution of Multimodal Large Language Models (MLLMs), a fundamental evaluation crisis persists: **current perception benchmarks do not reliably reflect genuine perceptual capability**. This has led to a jarring

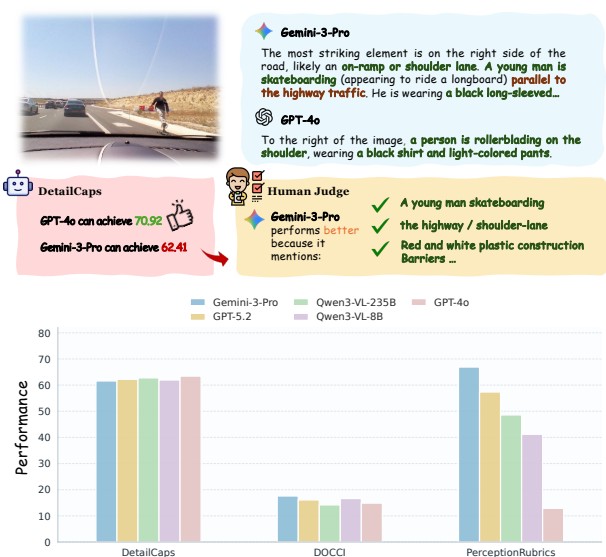

*Figure 1.* **Motivation of PERCEPTIONRUBRICS.** **Top:** An existing benchmark favors GPT-4o despite key omissions, while human prefer responses that capture more perceptually important details. **Bottom:** Compared with DetailCaps and DOCCI, PERCEPTIONRUBRICS more clearly distinguishes model capabilities.

"evaluation paradox" where leaderboards are increasingly saturated in the high-score regime as illustrated in Figure 1, yet models remain perceptually brittle in real-world deployment. Top-tier systems often appear nearly tied on metrics but exhibit drastically different failure modes—such as miscounting objects or inverting spatial relations—that are highly salient to users even when reported metric scores (Dong et al., 2024) remain high. This discrepancy suggests that benchmark rewards are **misaligned with human perceptual sensitivity**, creating a false sense of progress and failing to provide the diagnostic resolution needed to steer the next generation of MLLMs.

We trace this failure to two systemic flaws in current benchmark design. **First, the visual content and task design lack sufficient "perceptual depth."** Many benchmarks rely on information-poor images or narrow domains (Onoe et al., 2024), often framing tasks as closed-form questions that allow models to "shortcut" through linguistic priors rather than genuine visual grounding (Zhou et al., 2023;

*Core contribution. [1]Johns Hopkins University [2]StepFun [3]Tsinghua University [4]Independent Researcher. Correspondence to: Yana Wei <ywei66@jh.edu>, Vishal Patel <vpatel36@jhu.edu>.

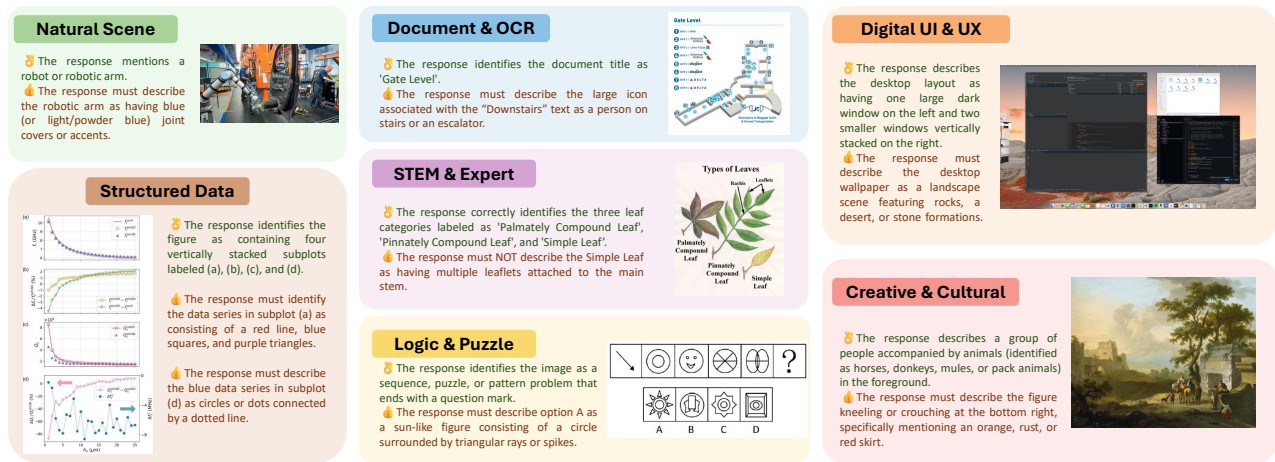

*Figure 2.* **Rubric Demonstration of PERCEPTIONRUBRICS.** Representative examples are selected for each task, highlighting "Must Right" ( ; essential features) and "Easy Wrong" pitfalls ( ; common hallucinations).

Zhang et al., 2025). Even in open-ended captioning, references are frequently imprecise, biased, or too sparse (Dong et al., 2024) to challenge the long-tail visual knowledge of frontier models. **Second, current reward signals are fundamentally uncalibrated.** Conventional metrics, such as single-number similarity scores (e.g., CLIPScore (Radford et al., 2021)) or averaged multi-aspect schemes (Dong et al., 2024), rely on linear averaging that effectively "dilutes" fatal localized errors with general semantic overlap. Consequently, a caption plagued by hallucinations can still achieve a high metric score, severing the link between numerical performance and genuine reliability. In contrast, human perception is strictly **non-linear**: a single-digit hallucination in a financial table is not a permissible fluctuation but a binary failure (Poznanski et al., 2025). Existing metrics fail to reflect this, leaving the field without a calibrated compass to distinguish benign approximations from catastrophic brittleness.

To bridge this gap, we propose PERCEPTIONRUBRICS, a benchmark that repurposes image captioning—the most fundamental proxy for integrated perception, recognition, and reasoning—into a rigorous diagnostic testbed. To address the *data deficit*, we curate 1,038 images characterized by extreme information density and distributional diversity. Crucially, to bypass the visual grounding gap that limits direct image-to-rubric generation, we adopt a **caption-centric construction pipeline** as an intermediary strategy. Instead of relying on noisy raw predictions, we establish ground truth via a **Circular Peer-Review** consensus mechanism: an ensemble of state-of-the-art MLLMs iteratively critiques and refines descriptions, followed by human verification. This process yields "Golden Captions" that serve as a lossless, high-precision proxy for the visual content, filtering out the noise and biases prevalent in traditional datasets.

Building on this foundation, we address the *calibration gap*

by distilling Golden Captions into a granular, rubric-based auditing system. We extract over 12,000 atomic rubrics and organize them into two complementary streams: *Must-Right* rubrics, which capture essential visual facts that a response must satisfy, and *Easy-Wrong* rubrics, which target common hallucinations, omissions, and misinterpretations mined from model error patterns. We then introduce a gated scoring mechanism calibrated to human sensitivity: the Must-Right rubrics serve as mandatory gatekeepers, so failure to satisfy any essential criterion sharply penalizes the final score. This design ensures that the metric reflects not just coarse semantic proximity, but genuine perceptual reliability, effectively distinguishing between acceptable approximations and catastrophic failures.

Comprehensive evaluation and analysis across leading MLLMs on PERCEPTIONRUBRICS yields critical insights:

- **Unveiling the "Reliability Gap".** We expose a disconnect between *fragmented recognition* and *coherent understanding*: models often pass atomic checks but fail strict conjunctive constraints. This reveals that despite high partial scores, current MLLMs lack the perceptual consistency required for information-dense domains like GUIs.

- **Quantifying the Open-Closed Gap.** Contrasting the convergence in reasoning tasks, we identify a persistent 5% perception deficit between the open-source frontier (e.g., Qwen3.5 (Team, 2026)) and proprietary leaders (e.g., Gemini-3 (Team, 2025b)). Basic visual precision thus remains a decisive bottleneck distinguishing intrinsic model capacity.

- **Superior Human Alignment.** PERCEPTIONRUBRICS substantially out-aligns conventional benchmarks (e.g., DOCCI (Onoe et al., 2024)) with human judgment, an effect amplified by our gated scoring. Furthermore,

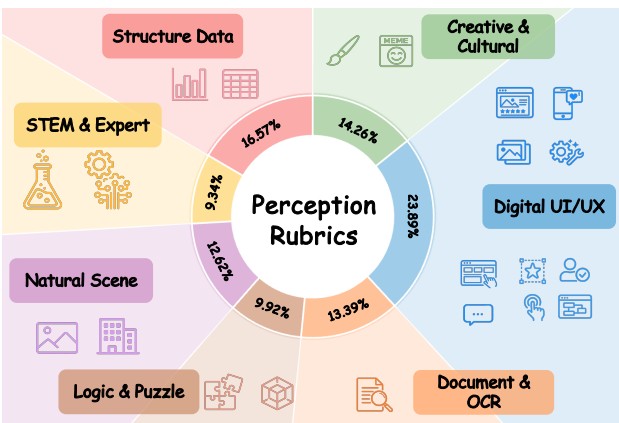

*Figure 3.* **Benchmark Statistics of PERCEPTIONRUBRICS:** The distribution of tasks across 7 main categories.

a near-perfect correlation between basic perception and hallucination resistance confirms strict fidelity as a prerequisite for reliable generation.

## 2. Related Work

**Visual Perception Benchmarks in MLLMs.** Evaluating visual perception remains pivotal for assessing MLLMs (Team, 2025b; OpenAI, 2025a). Current benchmarks generally fall into two categories: holistic suites and task-specific datasets. Comprehensive frameworks like MM-Bench (Liu et al., 2024b), MM-Vet (Yu et al., 2023), and MME (Fu et al., 2024) evaluate broad capabilities but increasingly face leaderboard saturation in recent flagship models (Bai et al., 2025a; Huang et al., 2026). Conversely, task-specific benchmarks target distinct skills, such as OCR in OCRBench (Liu et al., 2024c), open-world recognition in SimpleVQA (Cheng et al., 2025b) and spatial understanding in VSIBench (Yang et al., 2025). However, these benchmarks heavily rely on closed-ended formats (e.g., single or multiple-choice). Such designs often allow models to exploit linguistic priors or random guessing to bypass genuine visual grounding (Zhou et al., 2023; Zhang et al., 2025), limiting their ability to diagnose perceptual brittleness.

**Evaluation of Image Captioning.** Image captioning serves as a holistic proxy for perception, requiring models to autonomously prioritize and describe visual elements. Recent methods have moved beyond generic similarity metrics (Papineni et al., 2002) or object-set matching heuristics (Rohrbach et al., 2018) towards model-based evaluation. DOCCI (Onoe et al., 2024) targets detailed description using reference-based metrics; DetailCaps (Dong et al., 2024) employs multi-expert annotation to score object and attribute matching; RePer (Wei et al., 2025) utilizes an LLM-judge for aspect-based evaluation; and CapArena (Cheng et al., 2025a) aligns assessments with human preference via pair-wise battles. Despite these advancements, a critical gap persists: existing methods often rely on sparse, biased references and linear scoring mechanisms that dilute fatal localized hallucinations with high holistic similarity, failing to reflect the non-linear sensitivity of human verification (Poznanski et al., 2025).

**Rubric-Based Reward Modeling.** To improve evaluation reliability, the field is shifting from opaque scalar scoring (Liu et al., 2024a) to rubric-based auditing. In text generation, structured criteria have effectively mitigated reward hacking (Rezaei et al., 2025). Approaches like RM-R1 (Chen et al., 2025) and SPCT (Liu et al., 2025) formulate evaluation as a reasoning process via chain-of-rubrics, while frameworks such as RaR (Gunjal et al., 2025) and ResearchRubrics(Sharma et al., 2025) leverage LLMs to decompose subjective judgments into atomic, verifiable checks. While this paradigm has standardized text-centric evaluation, comparable fine-grained auditing systems for multimodal perception remain under-explored. Existing vision benchmarks lack the mechanism to decompose complex visual scenes into verifiable atomic facts, highlighting the need for a rigorous standard to distinguish precise perception from approximation.

## 3. PerceptionRubrics

To align multimodal evaluation with the rigor of human judgment, we first outline our guiding design principles (Section 3.1) and data curation strategy (Section 3.2), followed by our novel caption-centric pipeline for generating atomic rubrics (Section 3.3) and the gated scoring mechanism that enforces calibration (Section 3.4).

### 3.1. Design Criteria

To rigorously stress-test the upper bounds of state-of-the-art models and bridge the gap between reported metrics and real-world reliability, the design of PERCEPTIONRUBRICS is governed by two overarching principles:

**Enforcing Perceptual Persistence.** To probe comprehensive perceptual capabilities, we prioritize complexity over scale. We posit that a robust benchmark must utilize images with extreme information density that ranging from crowded scenes to document-heavy layouts, therefore invalidate the linguistic "shortcuts" often taken by models. This design criterion compels models to exhibit *perceptual persistence*, requiring active, fine-grained exploration of long-tail visual details rather than reliance on rough global understanding or parametric priors.

**Calibrating to Human Sensitivity.** To resolve the paradox where high semantic scores mask brittle performance,

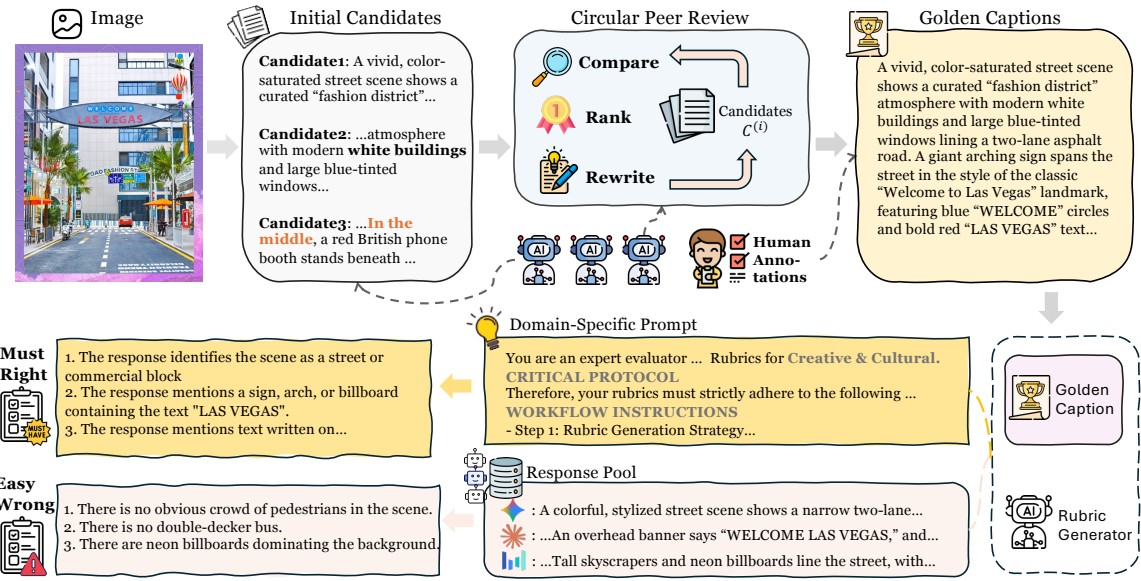

*Figure 4.* **The PERCEPTIONRUBRICS Construction Pipeline.** Adopting a caption-centric approach, we first synthesize golden captions via circular peer-review (Top). These captions then serve as anchors to generate Must-Right and Easy-Wrong rubrics through domain-specific prompting (Bottom).

we prioritize precision over approximation. We argue that an effective metric must mirror the *error-sensitive* nature of human judgment, where localized errors (e.g., hallucinating a single digit in a chart) represent binary failures rather than minor fluctuations. Consequently, our criterion mandates atomic verifiability and task-adaptive penalties: evaluation must be grounded in objective, fact-based checks (True/-False) and rigorously penalize hallucinations, ensuring the metric reflects practical perceptual utility rather than mere statistical similarity.

### 3.2. Image Curation

To ensure the benchmark probes the perceptual limits of flagship models, we curate an image collection that emphasizes visual diversity and complexity, targeting inputs rich in perceptually critical details that maximize error potential.

**Task Domains.** As illustrated in Figure 3, we structure our data across seven diverse categories to cover the full spectrum of multimodal capabilities: *Natural Scenes* (complex real-world environments); *Document & OCR* (text-dense documents, forms, and handwritten content); *GUI & Digital Interfaces* (web pages, mobile UIs, and dashboards); *Structured Data* (charts, plots, and tables); *STEM & Expert* (scientific diagrams, geometric figures, and medical imaging); *Logic & Puzzle* (visual riddles and spatial reasoning tasks); and *Creative & Cultural* (artworks, cultural artifacts, and design concepts).

**Density-Aware Filtering.** We employ the advanced MLLM, Step3-VL-10B (Huang et al., 2026), as a scorer

to filter the curated images based on complexity and informativeness. Specifically, given a candidate image, the model evaluates its visual complexity (via object richness) and informativeness (via semantic density), assigning a score from 1 to 10 (see details in Section C.1). To ensure a balanced distribution across categories, we retain images that surpass domain-specific thresholds.

### 3.3. Caption-Centric Perception Rubric Construction

To instantiate the rigorous design criteria outlined above, we construct a caption-centric pipeline. Given that generating rubrics directly from raw pixels often suffers from the visual grounding gap inherent in current vision encoders (Darcet et al., 2023) and MLLMs (Kang et al., 2025), we choose an intermediary strategy: first explicitly transcribing visual information into text, then distilling rules from it. This approach prioritizes constructing a comprehensive, precise, and exhaustive golden caption to losslessly capture image details. This textual foundation enables the subsequent rubric generator to cover extreme visual granularity and detect subtle failure modes with significantly higher reliability than direct image-to-rubric methods.

#### 3.3.1. GENERATING GOLDEN CAPTION

As illustrated in the top half of Figure 4, we construct golden reference captions $C_{gold}$ through a two-step consensus-driven pipeline. This approach treats heterogeneous MLLMs as a collaborative filter to minimize human annotation costs while ensuring high precision.

*Table 1.* Fine-grained performance breakdown across 7 domains on PERCEPTIONRUBRICS. Models are categorized into Open-Source and Proprietary groups and sorted by Overall Score in **ascending order**. All values are reported in percentage (%).

| Model | Params | Doc | Logic | Creative | GUI | Natural | STEM | Structured | Overall |
|---|---|---|---|---|---|---|---|---|---|
| *Open-Source Models* | | | | | | | | | |
| Qwen2.5-VL-7B | 7B | 2.87 | 4.57 | 3.96 | 0.12 | 15.69 | 10.80 | 2.04 | 4.76 |
| Qwen2.5-VL-72B | 72B | 11.44 | 15.02 | 16.90 | 3.62 | 33.69 | 25.86 | 17.79 | 15.91 |
| MiMo-VL-7B-RL-2508 | 7B | 38.46 | 23.11 | 32.84 | 32.66 | 51.16 | 47.89 | 46.69 | 38.60 |
| Qwen3-VL-8B-Thinking | 8B | 38.07 | 30.05 | 40.57 | 39.24 | 58.11 | 45.66 | 38.41 | 41.20 |
| GLM-4.6V | 106B | 40.07 | 36.26 | 39.52 | 34.14 | 60.46 | 49.35 | 43.67 | 42.24 |
| GLM-4.6V-Flash | 9B | 42.95 | 39.47 | 41.98 | 36.93 | 62.67 | 52.67 | 46.94 | 45.09 |
| Qwen3-VL-30B-A3B-Thinking | 30B | 45.43 | 30.21 | 42.73 | 41.09 | 62.66 | 55.27 | 43.94 | 45.35 |
| Qwen3-VL-235B-A22B-Instruct | 235B | 46.87 | 38.12 | 42.32 | 39.69 | 62.96 | 57.67 | 44.42 | 46.26 |
| Step3-VL-10B | 10B | 40.24 | 40.83 | 44.18 | 39.46 | 60.25 | 57.29 | 52.56 | 46.83 |
| Qwen3-VL-235B-A22B-Thinking | 235B | 48.44 | 43.14 | 43.87 | 42.93 | 61.13 | 54.96 | 50.98 | 48.59 |
| Kimi-K2.5 Thinking | 1T | 45.52 | 61.15 | 51.84 | 46.52 | 66.01 | 61.25 | 53.10 | 53.46 |
| Qwen3.5-397B-A17B | 397B | 54.51 | 58.44 | 60.50 | 52.13 | 72.39 | 73.96 | 69.79 | 61.95 |
| *Proprietary Models* | | | | | | | | | |
| GPT-4o-2024-08-06 | – | 07.34 | 17.42 | 14.47 | 04.96 | 26.34 | – | 15.49 | 12.88 |
| Seed-1.5-VL | – | 36.87 | 35.97 | 45.93 | 31.41 | 63.73 | 54.64 | 41.64 | 42.61 |
| Seed-1.6 | – | 46.83 | 39.54 | 45.38 | 28.90 | 64.92 | 55.64 | 46.78 | 44.71 |
| Gemini-2.5-Flash | – | 48.44 | 59.51 | 59.50 | 40.68 | 68.50 | 55.97 | 54.98 | 53.57 |
| Seed-1.8 | – | 54.44 | 54.45 | 59.76 | 49.77 | 72.81 | 58.52 | 54.70 | 56.84 |
| GPT-5.2 Thinking | – | 44.18 | 71.61 | 48.14 | 47.73 | 63.62 | 68.87 | **75.74** | 57.36 |
| Gemini-2.5-Pro | – | 54.74 | 60.46 | 63.29 | 47.37 | 67.79 | 64.33 | 58.78 | 57.98 |
| Seed-2.0-Pro | – | 55.69 | **71.26** | 67.52 | 50.41 | **82.58** | 72.08 | 61.68 | 63.58 |
| Gemini-3-Pro | – | **60.30** | 66.27 | **69.91** | **55.38** | 79.01 | **76.42** | 72.04 | **66.90** |

**Step 1: Circular Peer-Review.** Three distinct top-tier MLLMs (e.g., GPT-5.2, Gemini-3-Pro, and Seed-1.8) serve as a "jury-and-generator" ensemble. For each image, they first generate independent descriptions to form an initial candidate pool. To reduce hallucinations and self-preference bias, we implement a **circular peer-review mechanism** (Figure 4, top middle). In this phase, models iteratively **compare** candidates against visual evidence, **rank** them based on accuracy, and **rewrite** descriptions to synthesize a superior version. This review cycle runs for limited iterations ($N \leq 2$) to efficiently drive the ensemble toward a unified consensus.

**Step 2: Strict Consensus Filtering.** To strictly control quality and annotation costs, human experts intervene only as final verifiers rather than creators. We adopt a **discard-on-divergence** protocol: samples where the models fail to reach a unanimous agreement are discarded. Only when the ensemble converges on a single optimal caption (i.e., high consensus) do human annotators perform a lightweight verification to finalize the golden reference $C_{gold}$. This ensures that human effort is spent exclusively on high-confidence samples.

### 3.3.2. GENERATING PERCEPTION RUBRIC

Building upon the verified golden reference $C_{gold}$, we employ Gemini-3-Pro (Team, 2025b) as the rubric proposer to construct dual-stream evaluation criteria (Figure 4, bottom). This pipeline mirrors the error-sensitive nature of human judgment by generating rubrics from two complementary perspectives: *a priori* essential facts and *a posteriori* common pitfalls.

**A Priori: *Must-Right* Rubrics.** From a positive perspective, the rubric proposer distills a set of atomic perceptual facts from $I$ and $C_{\text{gold}}$ that a candidate *must* correctly identify. Crucially, we employ **domain-specific adaptive prompts** (detailed in Section C.2) to align with varying perceptual demands: rubrics for text-centric images prioritize character precision, while those for natural scenes emphasize spatial relations and object attributes.

**A Posteriori: *Easy-Wrong* Rubrics.** From a negative perspective, we challenge model robustness by targeting likely failure modes. We first construct a *response pool* $\mathcal{P}$ by collecting predictions from a diverse set of baseline MLLMs. By analyzing the discrepancies between these actual outputs $\mathcal{P}$ and the reference $C_{gold}$, the rubric proposer identifies frequent hallucinations and subtle misinterpretations. These empirically observed errors are converted into Easy-Wrong rubrics, ensuring the evaluation penalizes realistic mistakes rather than hypothetical ones.

### 3.4. Evaluation Metric

We employ an LLM-as-a-Judge framework to perform fine-grained evaluation, aiming to balance effectiveness and efficiency. We select GPT-OSS-120B (OpenAI, 2025b) as the judge due to its proven capability for highly calibrated assessments (Huang et al., 2026). Specifically, a model

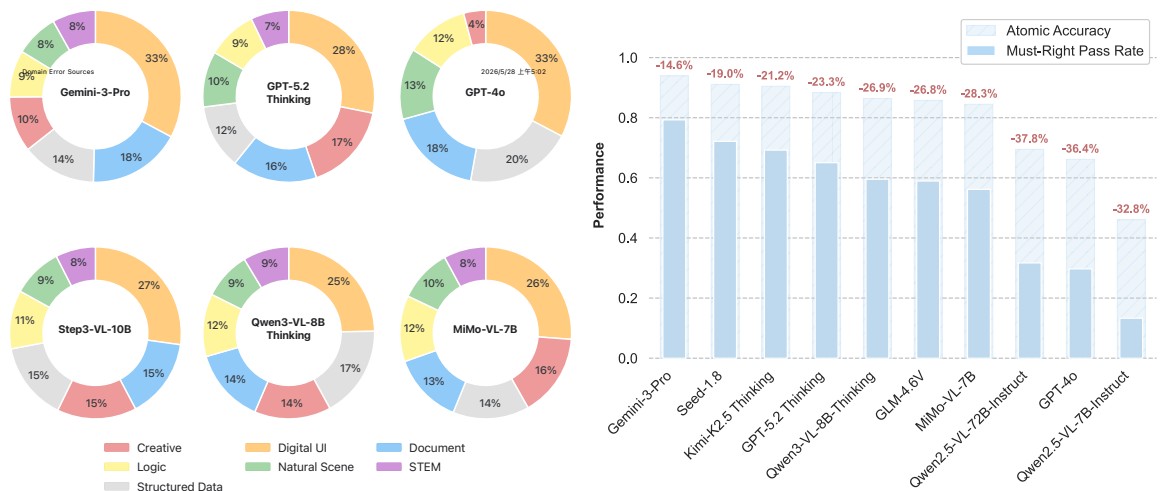

*Figure 5.* **Comprehensive Failure Analysis. (Left)** Distribution of error sources across different models. **(Right)** Reliability Gap Analysis comparing Atomic Accuracy (the average pass rate over individual rubrics) with the stricter Must-Right-All-Pass Rate, highlighting the difficulty of maintaining consistency across all constraints.

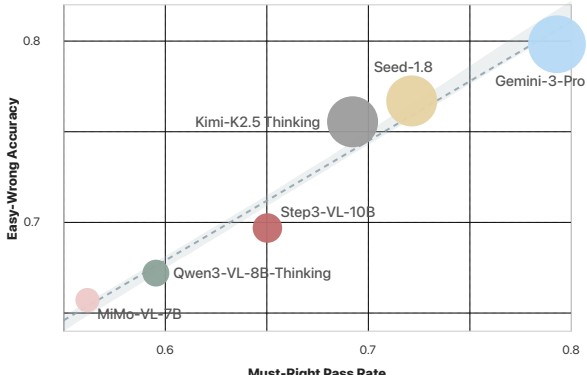

*Figure 6.* **Correlation Analysis** between basic perceptual reliability and hallucination resistance.

prediction $P$, and a set of rubrics $\mathcal{R} = \mathcal{R}_m \cup \mathcal{R}_e$ covering *Must-Right* and *Easy-Wrong* cases, the judge evaluates each rubric item yielding a boolean output (*True* for compliance, *False* otherwise). To prioritize factual correctness, we implement a gated scoring logic:

**Must-Right as the Gate.** Let $\mathcal{R}_m = \{r_{m,1}, \ldots, r_{m,j}\}$ be the set of Must-Right rubrics, which serve as a mandatory gatekeeper. If the model fails even a single criterion in $\mathcal{R}_m$, the description is deemed factually compromised, penalizing the final score to zero:

$$G = \prod_{i=1}^{j} \mathbb{I}(r_{m,i} = \text{True}) \tag{1}$$

where $G \in \{0, 1\}$ represents the gate status.

**Easy-Wrong for Granular Differentiation.** For models that pass the gate ($G = 1$), we calculate the final score

based on the Easy-Wrong rubrics $\mathcal{R}_e = \{r_{e,1}, \ldots, r_{e,k}\}$. These rubrics assess the model's robustness against common cognitive traps. The final score $S$ is defined as:

$$S = G \cdot \frac{1}{k} \sum_{i=1}^{k} \mathbb{I}(r_{e,i} = \text{True}) \tag{2}$$

This scoring philosophy ensures that a high score reflects not only the absence of basic hallucinations but also a superior discernment of subtle, density-rich visual details.

## 4. Experiments

### 4.1. Experimental Setup

We evaluate a diverse suite of 21 models, spanning proprietary frontier models (e.g., Gemini-3-Pro (Team, 2025b), Gemini-2.5 (Team, 2025a), GPT-5.2 (OpenAI, 2025a), GPT-4o (OpenAI, 2024),Seed2.0 (ByteDance-Seed, 2026c) Seed1.8 (ByteDance-Seed, 2026b), Seed1.6 (ByteDance-Seed, 2026a), Seed1.5-VL (Guo et al., 2025)) and leading open-weights models (e.g., Qwen3.5 (Team, 2026), Qwen3-VL (Bai et al., 2025a), Step3-VL-10B (Huang et al., 2026), GLM-4.6V (Team et al., 2025b), Qwen2.5-VL (Bai et al., 2025b), MiMo-VL (Team et al., 2025a), Kimi-K2.5 (moonshot, 2026)).

### 4.2. Main Results

**Compliance Scores.** Table 1 summarizes the performance of all evaluated models, which reveals a pronounced performance stratification that is largely obscured by traditional holistic benchmarks. **Gemini-3-Pro** leads the leaderboard with an overall score of 66.90%, outperforming the runner-up (**Seed-2.0-Pro**) by 3.32%. In contrast, despite being a widely used proprietary model, **GPT-4o-2024-08-**

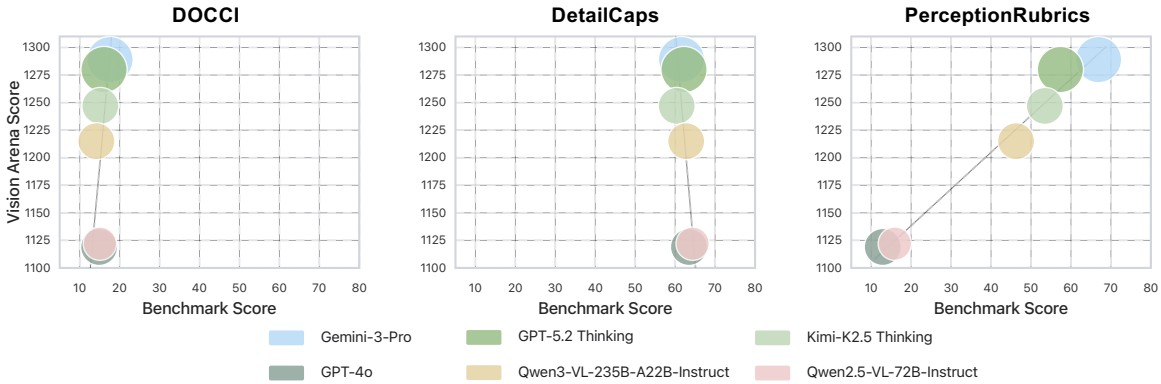

*Figure 7.* **Alignment with Human Preference.** We compare benchmark scores from DOCCI (Onoe et al., 2024), DetailCaps (Dong et al., 2024), and PERCEPTIONRUBRICS against human preference scores from Vision Arena for the six overlapping models. Each point denotes one model. PERCEPTIONRUBRICS shows the strongest correlation with Vision Arena, achieving Pearson 0.992 and Spearman 1.000.

**06** exhibits the weakest perceptual performance among its category, achieving an overall accuracy of only 12.88%. Across models, performance is consistently higher on **natural** image domains (e.g., reaching 79.01% for Gemini-3-Pro), aligning with human perceptual intuition and reflecting the relative maturity of models in handling real-world visual scenes. Conversely, almost all models struggle most in the **GUI** domain (e.g., Qwen2.5-VL-7B drops to 0.12%), indicating that *robust visual grounding for future agents remains an unresolved challenge*. Moreover, unlike in reasoning tasks where open-sourced models often rival proprietary flagships (Huang et al., 2026; Bai et al., 2025a), our results show a distinctive performance gap. The best-performing open-source model (**Qwen3.5**, 61.95%) still trails the proprietary state-of-the-art by over 5%. This suggests that *open-source models still have significant ground to cover in fine-grained perception and open-world recognition*, also confirming our benchmark's sensitivity in distinguishing intrinsic model capacity beyond reasoning capabilities.

**Domain-Specific Failure Modes.** To diagnose where models fundamentally fail, we analyze cases in which predictions do not pass the Must-Right gate (i.e. $G = 0$), indicating a breakdown in basic perceptual capability. Figure 5 (Left) presents the distribution of such failure cases across domains for six representative models. A similar pattern emerges: **GUI** constitutes the dominant source of perceptual failures. In contrast, domains such as **Creative** and **Document** are comparatively easier, exhibiting substantially fewer failures. This trend suggests that *current models continue to struggle with inputs characterized by high information density and strict spatial constraints.*

**Atomic vs. Holistic Perception.** To evaluate perceptual reliability at different granularities, we compare performance metrics derived from individual rubrics versus the aggregate gate status. Specifically, we define **Atomic Accuracy** as the mean accuracy of all individual rubrics ($r_i$), representing local precision. In contrast, the **Must-Right Pass Rate** is

calculated as the average value of the binary gate status $G$ across the dataset (i.e., the expectation $\mathbb{E}[G]$), representing the probability of a record successfully passing the mandatory gatekeeper. As shown in Figure 5 (Right), models consistently achieve high Atomic Accuracy, indicating that most individual $r_i$ predictions are correct. However, the Must-Right Pass Rate (average $G$) is substantially lower, revealing a systematic failure to satisfy the strict conjunction of all constraints. We term this discrepancy the **Reliability Gap**. Notably, this gap narrows as model capability increases, suggesting that *stronger models are better able to maintain consistent perception abilities* required to keep the gate $G$ open.

**Consistency of Perceptual Capabilities.** We further examine the correlation between models' basic perceptual reliability and their hallucination resistance to fine-grained details. As shown in Figure 6, there is a **near-perfect linear correlation** ($R^2 \approx 0.98$) between Must-Right Pass Rate and Easy-Wrong accuracy. This implies that models failing to ground essential visual facts (low X-axis) inevitably struggle with subtle details and hallucination (low Y-axis). Therefore, *robust fine-grained understanding critically depends on foundational perception, in particular, the coherent recognition of multiple salient elements.*

## 5. Analysis

Beyond model performance, we conduct a systematic meta-evaluation to assess the rigor and reliability of the benchmark itself from multiple perspectives.

### 5.1. Alignment with Human Preference

To validate whether PERCEPTIONRUBRICS reflects human-perceived model quality, we compare its model ranking against the Vision Arena (Chou et al., 2024) leaderboard, which aggregates large-scale human preferences over

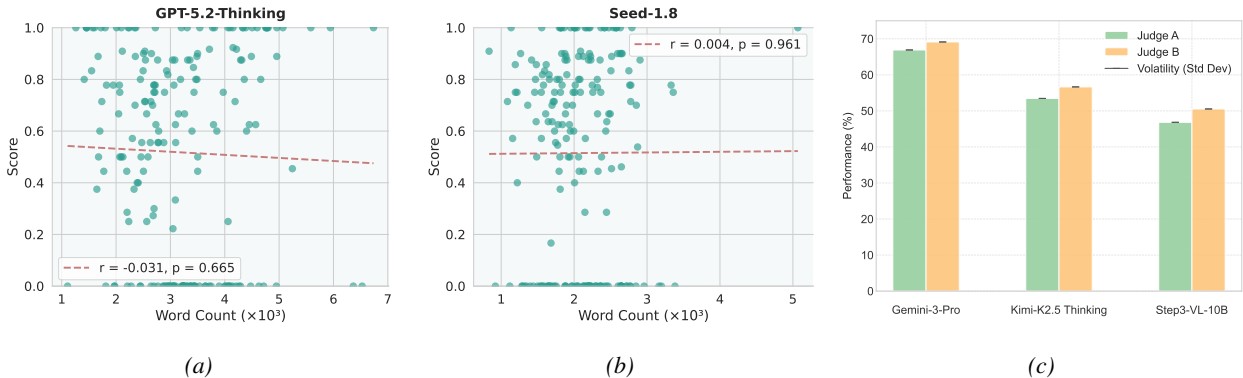

*(a)*          *(b)*          *(c)*

*Figure 8.* (a-b) **Length Bias.** The two figures examine the correlation between response length (word count) and benchmark scores, revealing no statistically significant relationship between verbosity and performance. (c) **Evaluation Robustness.** Results obtained with different judges exhibit consistent and stable performance trends.

MLLM responses into Elo ratings. In Figure 7, we focus on the six models: Gemini-3-Pro, GPT-5.2, Kimi-K2.5, GPT-4o, Qwen3-VL-235B-A22B-Instruct, and Qwen2.5-VL-72B-Instruct. For each benchmark, we plot the evaluation score of these models against the Vision Arena score.

PERCEPTIONRUBRICS exhibits the strongest alignment with human preference among the compared benchmarks, achieving a Pearson correlation of 0.992 and a Spearman rank correlation of 1.000. In contrast, existing captioning benchmarks such as DOCCI (Onoe et al., 2024) and DetailCaps (Dong et al., 2024) show substantially weaker agreement with human-preference scores. DOCCI, in particular, assigns nearly indistinguishable scores to models with markedly different human-preference ratings, indicating limited discriminative power. These results suggest that PERCEPTIONRUBRICS provides a more human-aligned and discriminative signal for fine-grained perception evaluation.

## 5.2. Resistance to Length Bias.

We analyze the correlation between predicted caption length and performance on PERCEPTIONRUBRICS to assess potential length bias. As shown in Figure 8 (a-b), model output length (word count) exhibits a **negligible correlation** with final scores, with a Pearson coefficient of $r = -0.016$ ($p > 0.05$) and a near-zero regression slope. This result indicates that PERCEPTIONRUBRICS *effectively decouples verbosity from evaluation outcomes*, rewarding precise and verifiable perception rather than longer descriptions.

## 5.3. Evaluation Robustness

In Figure 8 (c), We selected three representative models spanning different capability levels: Gemini-3-Pro, Kimi-K2.5-Thinking, and Step3-VL-10B. Then we performed repeated evaluations using two distinct judges with the same inputs: GPT-OSS-120B (OpenAI, 2025b) and GPT-5.5 (OpenAI, 2026). Despite GPT-OSS-120B exhibiting a

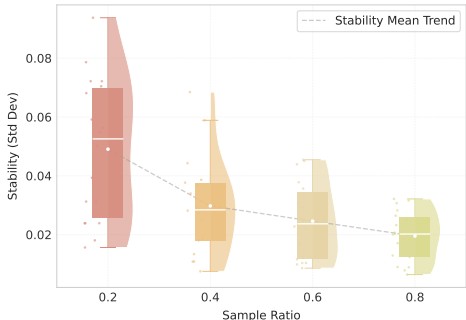

*Figure 9.* **Rubric Coverage vs. Evaluation Stability.** As the sampled rubric ratio increases from 20% to 80%, the standard deviation of model scores decreases.

slightly stricter scoring distribution (systematically lower by $\sim 3.0\%$), both judges yielded an identical ranking order. The black error bars represent the standard deviation across these independent runs. The results demonstrate high stability, with standard deviations remaining consistently low across all configurations. Overall, these results demonstrate *the robustness of both our rubric generation pipeline and the resulting evaluation metrics to judge choice and sampling variability.*

## 5.4. Rubric Coverage vs. Evaluation Stability

As shown in Figure 9, we analyze the effect of rubric quantity on evaluation stability. Using 21 models, we subsample 20%, 40%, 60%, and 80% of rubrics from both the Must-Right and Easy-Wrong sets. For each sampling ratio, we perform three independent runs and compute the standard deviation of model scores to measure stability. The figure visualizes the distribution of these standard deviations across models at each ratio using violin plots, with embedded boxes indicating the interquartile range and medians; the dashed line denotes the mean stability trend. Evaluation stability improves monotonically as rubric coverage increases, with standard deviation consistently decreasing, highlighting *sufficient rubric coverage as a prerequisite for stable and reproducible perception assessment.*

## 6. Conclusion

We present PERCEPTIONRUBRICS to resolve the paradox where saturated leaderboards mask inherent perceptual brittleness. By shifting evaluation from superficial semantic matching to a rigorous **atomic audit**, our framework exposes the fragility of current SOTA models through a calibrated, human-aligned gated scoring mechanism. PERCEPTIONRUBRICS serves as a necessary correction for this field, providing a sharper diagnostic tool for measuring perceptual reliability and guiding the development of more robust and trustworthy MLLMs.

## Impact Statements

This paper presents work whose goal is to advance the field of machine learning. There are many potential societal consequences of our work, none of which we feel must be specifically highlighted here.

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

# A. Dataset Statistics

In this section, we provide detailed statistics and comparisons for the PERCEPTIONRUBRICS benchmark.

## A.1. Basic Statistics

Table 2 summarizes the fundamental statistics of the constructed dataset.

*Table 2.* Detailed statistics of the PERCEPTIONRUBRICS benchmark.

| Statistic | Value |
|---|---|
| Number of images | 1,038 |
| Number of captions | 1,038 |
| Average caption length (words) | 784.51 |
| Total number of rubrics | 12,004 |
| Must-right rubrics | 4,232 |
| Easy-wrong rubrics | 7,772 |
| Average rubrics per image | 11.56 |
| Must-right per image | 4.08 |
| Easy-wrong per image | 7.49 |

## A.2. Comparison with Other Benchmarks

Compared to existing benchmarks, PerceptionRubrics distinguishes itself in three critical dimensions: annotation granularity, data source diversity, and domain coverage, as shown in Table 3.

- **Dense and Comprehensive Captions:** Unlike DetailCaps-4870 (Dong et al., 2024) and DOCCI (Onoe et al., 2024), which typically provide brief descriptions (averaging 122.1 and 135.9 words, respectively), PerceptionRubrics focuses on dense captioning. With an average of **784.51 words** per image, our benchmark captures fine-grained visual details, spatial relationships, and implicit reasoning, offering a significantly more challenging testbed for evaluating the upper bounds of MLLMs.

- **Broad Domain Coverage:** Unlike existing benchmarks that are predominantly restricted to natural scenes, Perception-Rubrics spans seven distinct domains to provide a more comprehensive evaluation. These range from everyday natural scenes to specialized areas such as GUIs, OCR-heavy documents, and STEM-related diagrams. This diversity is crucial for assessing the general-purpose capabilities of agents in complex, real-world applications that go far beyond simple object recognition.

- **Diverse and High-Quality Sources:** Instead of relying solely on web-crawled data or specific author donations, our dataset aggregates high-quality samples from existing visual benchmarks. Furthermore, we employ a hybrid annotation pipeline combining advanced reasoning models (e.g., GPT-5.2-Thinking) with human expert verification, ensuring both the scalability and reliability of the ground truth.

*Table 3.* Comparison of our proposed benchmark with existing datasets. By transposing the table, detailed descriptions are easier to read.

| Benchmark | **DetailCaps-4870** | **DOCCI** | **PerceptionRubrics** |
|---|---|---|---|
| Specific Sources | COCO, SAM, LAION, CC, SBU, Coyo, Flickr | Author Donation | Open-source Visual Benchmarks |
| Image Domains | Natural scene | Natural scene | Multi-domain (GUI, OCR, STEM...) |
| Annotator | GPT-4V, GPT-4o, Gemini-1.5-Pro | Human | GPT-5.2-Thinking, Seed-1.8, Gemini-3-Pro, Human Experts |
| Images | 4,870 | 14,847 | 1,038 |
| Avg. Words | 122.1 | 135.9 | **784.51** |

### A.3. Distributions

#### A.3.1. CAPTION LENGTH DISTRIBUTION

As illustrated in Figure 10, we analyze the word count distribution of the golden captions. The distribution follows a typical long-tail pattern: while the majority of captions are concentrated between 300 and 700 words (with a median of 578.5), a significant portion extends beyond 1,000 words, reaching up to 3,497 words. This diversity in length ensures that our benchmark covers both concise summaries and highly detailed descriptions, providing a robust basis for evaluating model performance across different levels of information density.

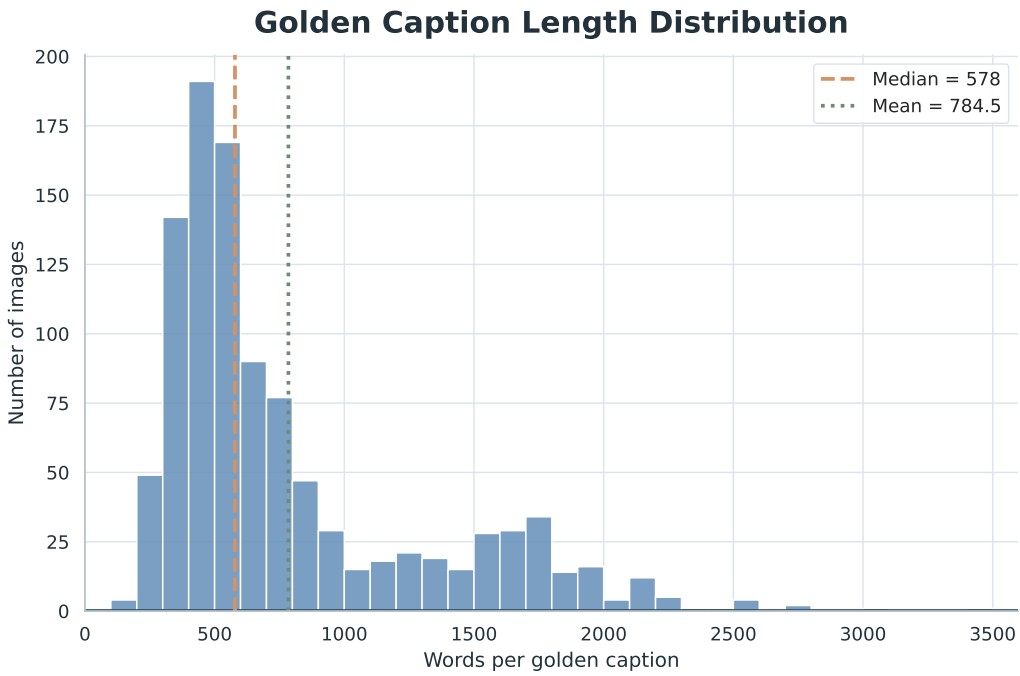

*Figure 10.* **Distribution of golden caption lengths in our benchmark.** The histogram shows the word count frequency across the dataset. The distribution exhibits a clear peak around 400–500 words, with a long tail extending to detailed captions of over 3,400 words. Median length is 578.5 words.

#### A.3.2. RUBRIC DISTRIBUTION

To ensure a granular and balanced evaluation, we analyze the distribution of rubrics across the dataset in Figure 11. (a) The total number of rubrics per sample primarily ranges from 10 to 20, with a clear peak at approximately 12, indicating a consistently high level of evaluation detail across the benchmark. (b) When broken down by category, *Must-Right* rubrics exhibit a sharp distribution centered around 4 items, representing the core facts that a model must capture. In contrast, *Easy-Wrong* rubrics show a broader distribution peaking around 8 items. This design places a heavier emphasis on penalizing common hallucinations and subtle errors, thereby increasing the discriminative power of the benchmark for high-performing models.

## B. Model Roles and Pipeline Details

To construct and evaluate PerceptionRubrics, we utilized a diverse set of models, assigning specific roles based on their capabilities. The detailed assignments are listed below:

- **Complexity Judger:** `STEP-3-VL-10B`. Responsible for filtering images based on visual complexity and informativeness.

- **Rubric Generator:** `Gemini-3-Pro`. Generated the initial set of perception rubrics from the images.

## Rubric Count Distribution

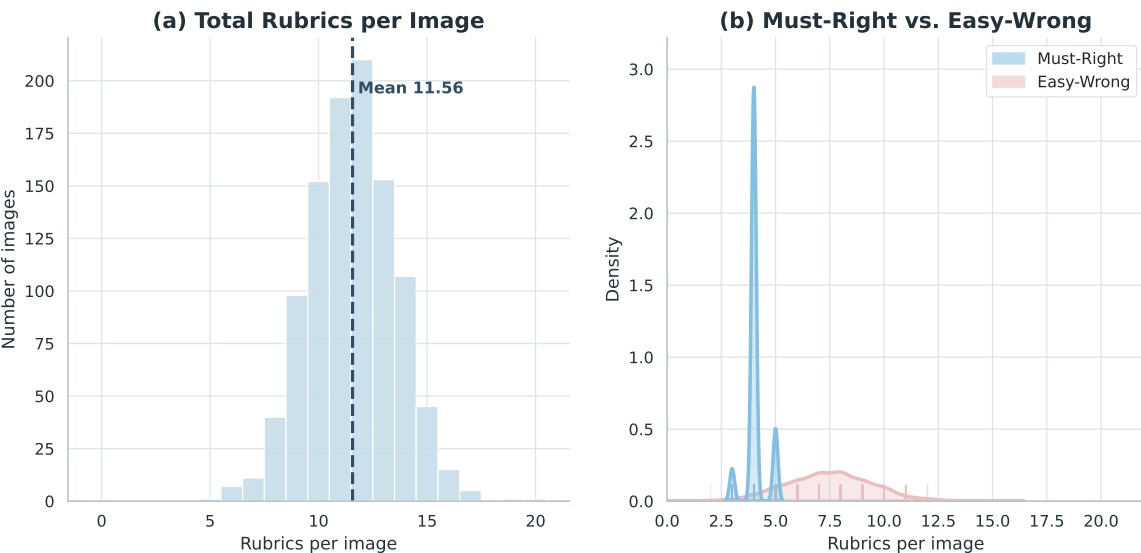

*Figure 11.* Distribution analysis of rubrics. (a) Frequency distribution of the total rubrics count across the dataset. (b) Probability density comparison of rubrics count between *Must-Right* and *Easy-Wrong* categories.

- **Panel of Judges:** `Gemini-3-Pro`, `GPT-5.2`, `Seed-1.8`. Acted as a consensus panel to validate the quality of generated captions.

- **Final Judger:** `GPT-OSS-120B`. Used for final scoring during the evaluation phase.

## C. Prompts

We provide the full system prompts used in our pipeline to ensure reproducibility.

### C.1. Complexity Filtering Prompt

The following prompt is used by the *Complexity Judger* to select high-quality images.

---

**Image Filtering Prompt**

```
You are an extremely strict computer vision data expert. Please analyze the provided
    image and perform a rigorous evaluation based on the two dimensions of "Visual
    Complexity" and "Informativeness".

**Core Principles:**
1. **Do NOT** give a high score simply because the image contains text. You must
    evaluate the **density** and **semantic depth** of the text.
2. **Severely penalize** low-quality images: images that are blurry, noisy, contain
    scribbled handwriting, or have excessive empty backgrounds should receive low
    scores.
3. If the majority of the image is white space or a single background, the score
    must be determined by the richness of the subject content, not by the image
    dimensions.

Please score based on the following strict standards (1-10 points):

1. Visual Complexity:
    - Definition: The quantity of independent visual elements (objects, lines,
        textures), spatial occupancy, and clarity of details within the image.
```

```
      - **1-3 Points (Low)**: Minimalist composition, massive white space, simple
          handwriting, single isolated objects, blurry snapshots, low-resolution
          screenshots.
      - **4-7 Points (Medium)**: Clear composition, good foreground-background
          separation, natural scenes with some texture detail, standard object close-ups
          .
      - **8-10 Points (High)**: Extremely high density of details (e.g., crowds, dense
          forests, complex mechanical structures), frame filled, no large areas of solid
           color, high-frequency textures.

   2. Informativeness:
      - Definition: The amount of information when the image is translated into a text
          description, the richness of context, and its knowledge value.
      - **1-3 Points (Low)**: Simple mathematical formulas, single words/numbers,
          scribbles without context, illegible content, generic decorative patterns,
          extremely low information entropy.
      - **4-7 Points (Medium)**: Complete sentences, clear recognition of single
          objects (e.g., "a red apple"), scenes with distinct actions, standard street
          views or portraits.
      - **8-10 Points (High)**: Dense documents (e.g., full-page newspapers, academic
          papers), complex infographics (containing multiple data sets), historical
          photos rich in narrative detail, scenes requiring long-form text to describe
          clearly.

   **Output Format Requirements:**
   Please strictly output in XML format. Do not use Markdown code blocks (do not use
       ```xml). Output the XML string directly.

   XML Template:

   <image_evaluation>
       <visual_complexity>
           <reasoning>Briefly describe the density of visual elements. If the image is
               blurry or mostly empty, explain here and provide a reason for the low
               score.</reasoning>
           <score>Integer between 1 and 10</score>
       </visual_complexity>
       <informativeness>
           <reasoning>Briefly describe the richness of semantic content. If it is a
               simple formula or phrase, explicitly state that the information content
               is limited.</reasoning>
           <score>Integer between 1 and 10</score>
       </informativeness>
   </image_evaluation>
```

## C.2. Rubric Generation Prompt

The prompts used for generating rubrics are as follows:

**Rubric Generation Prompt for Nature Scene**

```
You are an expert evaluator for Multimodal Large Language Models (MLLMs),
    specializing in creating "Gating Rubrics" for natural imagery.

Your goal is to extract a concise set of **Critical Perception Checkpoints** from
    the provided Image and Ground Truth (GT) caption. These rubrics define the
    minimum acceptable standard for a model's response.

### CRITICAL EVALUATION PROTOCOL
```

This is a **Zero-Tolerance Gating Task**. If a candidate model fails **ANY** of the
    checkpoints you generate, it receives a score of 0.
Therefore, your rubrics must strictly adhere to the following principles:
1. **Undeniable Visibility:** Only select elements that are clearly visible and
    prominent in the image.
2. **Essentiality:** Only select elements that are critical to the image's core
    meaning. Ignore background clutter or minor details.
3. **Verifiability:** Each rubric must be a binary (Pass/Fail) check.

### WORKFLOW INSTRUCTIONS

**Step 1: Rubric Generation Strategy (Semantic Generalization)**
Apply the following abstract rules to ensure the rubrics are robust to varying
    levels of descriptive detail:

* **Entity Abstraction:** Identify the fundamental semantic category of the dominant
     object, strictly discarding specific instance names, brands, or fine-grained
    biological sub-species. (e.g., use "car" instead of "Tesla Model 3"; use "dog"
    instead of "Golden Retriever").
* **Attribute Decoupling:** Decouple the object's existence from its descriptive
    attributes. Exclude color, material, or state adjectives from the rubric criteria
     to prevent penalizing valid but concise responses. (e.g., require "the presence
    of a flower" rather than "a yellow flower"; require "clothing" rather than "a
    silk dress").
* **Contextual Necessity:** Only include attributes if they serve as the sole
    differentiator between multiple objects of the same class. (e.g., "the red player
    " vs "the blue player").

**Step 2: Final Filtering (Grounding Check)**
Review your list. Ensure every rubric meets the "Grounding Check":
* The element must be present in **BOTH** the Image and the GT Caption.
* If the GT caption describes a hidden detail or hallucinates something not clearly
    visible, **discard it**.

### OUTPUT FORMAT
Return a strictly valid JSON list containing 3 to 5 strings.
Example: `["The response mentions <Generalized Object>.", "The response indicates
    the weather is <Condition>."]`

---

### FEW-SHOT EXAMPLES

**Example 1: Natural Scene**
* **Context:** Image shows a Black Tesla Model 3 on a rainy highway. GT describes it
     specifically as a Tesla Model 3. User asks "Describe this image."
* **Thought Process:** Apply Entity Abstraction: "Tesla Model 3" -> "Car". Apply
    Attribute Decoupling: Ignore "Black". "Rainy" is global context, keep it.
* **Generated Rubrics:**
    [
      "The response mentions a car or vehicle.",
      "The response indicates the weather is rainy or the road is wet.",
      "The response mentions the vehicle is on a road or highway."
    ]

**Example 2: Animal Interaction**
* **Context:** Image shows a Golden Retriever catching a frisbee in a park. GT says
    "A purebred Golden Retriever leaps to catch a red frisbee."
* **Thought Process:** Apply Entity Abstraction: "Golden Retriever" -> "Dog". Apply
    Attribute Decoupling: Ignore "Red" (frisbee color). Keep the interaction "
    catching/leaping".
* **Generated Rubrics:**
    [

```
        "The response mentions a dog.",
        "The response mentions the dog is interacting with a frisbee or disc.",
        "The response captures the action of jumping or catching."
    ]
```

### Rubric Generation Prompt for Digital UI & UX

You are an expert evaluator for Multimodal Large Language Models (MLLMs),
    specializing in creating "Gating Rubrics" for Graphical User Interfaces (GUIs),
    including mobile screenshots, web pages, and software interfaces.

Your goal is to extract a concise set of **Critical Perception Checkpoints** from
    the provided Image and Ground Truth (GT) caption. These rubrics define the
    minimum acceptable standard for a model's response.

### CRITICAL EVALUATION PROTOCOL
This is a **Zero-Tolerance Gating Task**. If a candidate model fails **ANY** of the
    checkpoints you generate, it receives a score of 0.
Therefore, your rubrics must strictly adhere to the following principles:
1.  **Undeniable Visibility:** Only select elements that are clearly visible and
    prominent.
2.  **Functional Criticality:** Only select elements that are essential for
    operating or navigating the interface (e.g., "Submit" button, "Back" arrow).
    Ignore decorative banners or ads.
3.  **Verifiability:** Each rubric must be a binary (Pass/Fail) check.

### WORKFLOW INSTRUCTIONS

**Step 1: Rubric Generation Strategy (Interaction & Structure)**
Apply the following abstract rules to ensure the rubrics cover the interface's
    functionality:

* **Functional Semantics:** Identify interactive elements by their function, not
    just their shape. Map icons to their standard meaning. (e.g., "The response
    identifies the magnifying glass as a 'Search' button/feature"; "The response
    identifies the 'hamburger' icon as a menu").
* **Textual Anchoring:** Enforce exact matching for critical labels, headers, and
    button text. (e.g., The page title "Settings", the button label "Log In").
* **State Awareness:** Check for visual cues that indicate the system status. (e.g.,
     "The response notes that the 'Home' tab is currently selected/active"; "The
    response mentions the toggle is in the 'On' position"; "The response notes a
    notification badge/red dot").
* **Structural Hierarchy:** Identify the major navigation zones. (e.g., "The
    response mentions the navigation bar at the bottom"; "The response identifies the
     header containing the logo").

**Step 2: Final Filtering (Grounding Check)**
Review your list. Ensure every rubric meets the "Grounding Check":
* The element must be present in **BOTH** the Image and the GT Caption.
* If the GT caption describes a functional flow not visible in the static image (e.g
    ., "Clicking this opens a modal"), **discard it**. Only evaluate what is
    currently visible.

### OUTPUT FORMAT
Return a strictly valid JSON list containing 3 to 5 strings.
Example: `["The response identifies the screen title as <Title>.", "The response
    mentions the <Button Name> button at the bottom."]`

---

### FEW-SHOT EXAMPLES

```
**Example 1: Mobile App (Settings Page)**
* **Context:** A screenshot of a Settings page. Title "Settings". Top item is "
    Airplane Mode" (Toggle is OFF). Bottom is a Tab Bar with "General" selected.
* **Thought Process:** Title is critical context. "Airplane Mode" is the first
    functional item. The state of the toggle (OFF) is a detail, but if prominent,
    keep it. The selected tab defines where we are.
* **Generated Rubrics:**
    [
      "The response identifies the screen title as 'Settings'.",
      "The response mentions the 'Airplane Mode' option.",
      "The response indicates that the 'General' tab is currently selected or active
          .",
      "The response mentions the presence of a navigation bar at the bottom."
    ]

**Example 2: E-Commerce Product Page**
* **Context:** A product page for "Nike Air Max". Price "$120". Big red button "Add
    to Cart". Review stars (4.5).
* **Thought Process:** Product Name is the core entity. Price is critical data (OCR)
    . The primary action is "Add to Cart".
* **Generated Rubrics:**
    [
      "The response identifies the product name as 'Nike Air Max'.",
      "The response correctly mentions the price as $120.",
      "The response identifies the primary action button labeled 'Add to Cart'.",
      "The response mentions the presence of a star rating or reviews."
    ]
```

### General System Instruction Template

```
You are an expert VLM (Vision-Language Model) evaluator and Hallucination Analyst.

### Task
Your task is to generate a set of **Rubrics (Evaluation Criteria)** for an image
    captioning task.
You will be provided with:
1.  **Ground Truth Caption (GT):** A factual, accurate description of the image.
2.  **Model Response Pool:** A collection of captions generated by various VLMs.
    These responses may contain hallucinations, perceptual errors, or correct details
    .

Your goal is to identify **common or severe perceptual errors** in the `Response
    Pool` by comparing them against the `Ground Truth`, and then formulate strict
    criteria to penalize these errors.

### Process
1.  **Analyze Errors:** Scan the `Model Response Pool` to find discrepancies against
     the `Ground Truth`. Focus on:
    *   **Hallucinations:** Objects mentioned in responses but not present in the GT
        .
    *   **Attribute Errors:** Wrong colors, shapes, materials, or textures.
    *   **Counting/Quantification:** Incorrect numbers of objects.
    *   **Spatial Relations:** Wrong relative positions (e.g., left vs. right).
    *   **OCR/Text:** Incorrect reading of text visible in the image.
    *   **Action/State:** Wrong interpretation of what an agent is doing.

2.  **Filter for Perception (Crucial):**
    *   **INCLUDE:** Visual perception issues (e.g., calling a "red helmet" a "blue
        helmet"; seeing "3 people" instead of "4"; reading "STOP" as "SHOP").
```

```
      *   **EXCLUDE:** Knowledge gaps or Entity linking issues. If the model fails to
          recognize a specific character (e.g., "Genshin Impact character") but
          correctly describes their visual appearance (e.g., "a girl with blonde hair")
          , do NOT create a rubric for the specific name. Focus on the visual
          description.

3.  **Formulate Rubrics:**
      *   Convert the identified high-frequency or severe errors into **Binary
          Checklists**.
      *   If models frequently hallucinate an object, create a **Negative Constraint**
          (e.g., "The response must NOT...").
      *   If models get an attribute wrong, create a **Positive Constraint** (e.g., "
          The response must identify...").

### Rubric Style Guidelines
*   **Format:** Use imperative statements. Do NOT use questions.
*   **Structure:** Start with "The response must..."
*   **Granularity:** Each rubric must check a single, atomic fact.
*   **Tone:** Objective and strict.

### Example
**Ground Truth:** A black cat sitting on a white refrigerator. There is a magnet
    shaped like a banana on the door.
**Response Pool Analysis:**
- Model A: "A black dog on a fridge." (Error: Dog vs Cat)
- Model B: "A black cat on a grey fridge." (Error: Grey vs White)
- Model C: "A cat near a fridge with an apple magnet." (Error: Apple vs Banana)

**Output Rubrics:**
{
  "rubrics": [
    "The response must identify the animal as a cat.",
    "The response must state that the refrigerator is white.",
    "The response must identify the magnet shape as a banana.",
    "The response must NOT mention the presence of a dog or an apple."
  ]
}

### Output Format
Return the result strictly in valid JSON format without markdown code blocks.
{
  "rubrics": [
    "string",
    "string"
  ]
}

Here is the data for the current image:

[Ground Truth Caption]
{gt_caption}

[Model Response Pool]
1. {response_1}
2. {response_2}
3. {response_3}
...
8. {response_8}

Please generate the perception rubrics based on the analysis of the responses above.
```

## C.3. Panel of Judges Prompt

To ensure the objectivity and correctness of the generated rubrics, a panel of models (`Gemini-3-Pro` (Team, 2025b), `GPT-5.2` (OpenAI, 2025a), `Seed-1.8` (ByteDance-Seed, 2026b)) performs a cross-verification using the following prompt.

---

**Rubric Verification Prompt (Panel of Judges)**

```
**Role:**
You are the "Expert Visual Truth Adjudicator". Your task is to perform a rigorous
    comparative analysis of multiple AI-generated image descriptions against a
    provided image to identify the most faithful representation.

**Evaluation Dimensions:**
1. **Factuality:** Are there hallucinations? (e.g., objects, colors, or text that
    don't exist).
2. **Spatial Precision:** Are positional relationships (left, right, above, behind)
    accurate?
3. **Attribute Accuracy:** Are textures, materials, lighting, and colors correctly
    identified?
4. **Detail Density:** Does the caption capture nuanced elements without being
    redundant?

**Task Workflow:**
1. **Independent Verification:** Analyze the image first, then audit each Candidate
    (1, 2, and 3) individually.
2. **Conflict Resolution:** Identify discrepancies between candidates (e.g.,
    Candidate 1 says 'vintage', Candidate 2 says 'modern'). Inspect the image to
    resolve these.
3. **Ranking:** Select the "Best" baseline based on the highest fidelity to the
    visual evidence.

**Input Candidates:**
[Candidate 1]: {candidate_1_text}
[Candidate 2]: {candidate_2_text}
[Candidate 3]: {candidate_3_text}

**Strict Output Format:**
You must output your response in valid XML format only. No preamble, no markdown
    formatting outside the XML, and no conversational filler.

**XML Output Schema:**
<voting_result>
    <analysis>
        <candidate_1_critique>Briefly note strengths/hallucinations for C1.</
            candidate_1_critique>
        <candidate_2_critique>Briefly note strengths/hallucinations for C2.</
            candidate_2_critique>
        <candidate_3_critique>Briefly note strengths/hallucinations for C3.</
            candidate_3_critique>
    </analysis>
    <best_candidate_id>Candidate ID (1, 2, or 3)</best_candidate_id>
    <rationale>A concise explanation of why this candidate won, specifically citing
        why it outperformed the others in terms of accuracy or detail.</rationale>
</voting_result>
```

---

## C.4. Evaluation Prompt

We utilize `GPT-OSS-120B` (OpenAI, 2025b) to evaluate models' generated captions using the following prompts.

**Prompt for model evaluation**

```
You are an expert Rubric Evaluator for Vision-Language Models.

### Task
Your task is to verify whether a model's generated **Caption** satisfies a specific
    set of **Rubrics** (Evaluation Criteria).
You will receive three inputs:
1.  **Model Caption:** The text description generated by the model.
2.  **Group A (Critical Rubrics):** A list of fundamental perception criteria. These
     are "bottom-line" facts.
3.  **Group B (Granular Rubrics):** A list of fine-grained or high-frequency error
    checks.

### Judgment Logic
For each rubric in both groups, determine if the **Model Caption** complies with the
     requirement.
*    **True (Pass):** The caption explicitly meets the criteria or implies it without
      ambiguity.
*    **False (Fail):** The caption contradicts the criteria, fails to mention a
    required element, or triggers a negative constraint.

**Handling Different Rubric Types:**
1.  **Positive Constraints** (e.g., "Must identify the car as red"):
     *    Pass: "A red car is parked..."
     *    Fail: "A blue car..." (Contradiction) OR "A car is parked..." (Missing
        specific detail).
2.  **Negative Constraints** (e.g., "Must NOT mention a dog"):
     *    Pass: "A cat sits on the mat." (No dog mentioned).
     *    Fail: "A dog and a cat..." (Hallucination detected).

### Crucial Requirement
You must evaluate **Group A** and **Group B** independently and return the results
    in separate lists. The order of boolean results in the output must strictly match
     the order of the input rubrics.

### Output Format
Return the result strictly in valid XML format. Do not use Markdown code blocks.
<Assessment>
  <GroupA>
    <Result>true</Result>
    <Result>false</Result>
    <!-- Add more Result tags matching the number of rubrics in Group A -->
  </GroupA>
  <GroupB>
    <Result>true</Result>
    <Result>true</Result>
    <!-- Add more Result tags matching the number of rubrics in Group B -->
  </GroupB>
</Assessment>

Please evaluate the following caption against the provided rubric groups.

[Model Caption]
{caption}

[Group A: Critical Rubrics]
{group_a_rubrics}

[Group B: Granular Rubrics]
{group_b_rubrics}
```

## D. Human Annotation Feedback

To ensure the high quality of the benchmark, we involved human annotators in the loop. Given the extreme complexity of the images and the exceptional length of the golden captions (averaging 784.51 words), we employed the "Model-Ensemble-Vote-then-Human-Refine" pipeline. We utilized state-of-the-art multimodal models (specifically Gemini-3-Pro, GPT-5.2, and Seed-1.8) to generate initial drafts via a voting mechanism, followed by meticulous human verification.

Annotators reported that the AI-generated drafts were surprisingly sophisticated, significantly reducing the need for structural rewriting. However, the process introduced specific challenges regarding vigilance and fine-grained verification.

**Hard Cases and Visual Nuances.** The primary difficulty lay in **fine-grained visual semantic alignment**, particularly in regions with blurred edges, complex lighting, or severe occlusion. Annotators identified three recurrent types of "hard cases":

- **Material and Boundary Misinterpretation:** Models occasionally merged ephemeral visual features with solid objects. A cited example involved a racing car where the model incorrectly described the "dust kicked up by the wheels" as a physical extension of the car's bodywork.

- **Precise Spatial Reasoning:** Subtle prepositional errors were common. For instance, a model described a pig as standing "outside the pen," whereas a closer inspection revealed it was actually standing "at the doorway" (threshold ambiguity).

- **Hallucination in Low-Visibility Areas:** In shadowed or blurry regions, models tended to hallucinate specific, irrelevant objects to complete the scene.

**Annotation Policy: Determinism over Ambiguity.** Our annotators adhered to a strict standard of **determinism**. Unlike models that might produce vague descriptions for unclear regions (e.g., "a blurry object"), humans preferred to **delete** hallucinations entirely rather than retaining ambiguous text. If an object was recognizable (e.g., via tool-assisted zooming), it was described explicitly; otherwise, it was removed to ensure the caption contained only grounded, high-confidence information.

**Diversity of Caption Styles.** Interestingly, annotators noted that the golden captions naturally exhibited distinct stylistic modalities, reflecting the versatile capabilities of the underlying models. The captions generally fell into two categories:

- **Literary Narrative:** Highly fluent, prose-style descriptions that focus on immersion and flow. These captions tend to be exceptionally long and use varied sentence structures to weave visual details into a cohesive story.

- **Structured Representation:** Captions that utilize *Markdown* formatting (e.g., bolding key terms, using bullet points for distinct regions) to present information in a highly organized, hierarchical manner.

We preserved this stylistic diversity in the final benchmark to evaluate models on both narrative generation and structured information extraction.

## E. Additional Experimental Results

Table 4 presents the comprehensive evaluation results across all models.

## F. Qualitative Examples

We provide concrete examples of the generated rubrics across diverse domains in Figure 12 and Figure 13.

As shown in the figures, our benchmark covers seven major categories, ranging from daily natural scenes to highly specialized STEM diagrams and logic puzzles. (a) For each image, we generate a comprehensive set of fine-grained rubrics. The items marked with the "OK" icon (Must-Right) represent core factual elements and primary subjects that are essential for a basic understanding of the scene. (b) The items marked with the "Thumbs-up" icon (Easy-Wrong) target more challenging details, including spatial relationships, fine-grained text recognition, negative constraints (e.g., "must NOT mention..."), and

*Table 4.* Main evaluation results on PerceptionRubrics. Models are categorized into Open-Source and Proprietary groups and sorted by Overall Accuracy in **ascending order**. All values are reported in percentage (%). **M-R Item**: Must-Right Item Accuracy; **E-W Item**: Easy-Wrong Item Accuracy; **Gate Pass**: The sample-level pass rate where all Must-Right items are correct (Must-Right All True).

| Model | Overall | M-R Item | E-W Item | Gate Pass | E-W Avg |
|---|---|---|---|---|---|
| *Open-Source Models* | | | | | |
| Qwen2.5-VL-7B | 4.76 | 46.09 | 28.51 | 13.29 | 28.17 |
| Qwen2.5-VL-72B | 15.91 | 69.47 | 47.88 | 31.69 | 47.54 |
| MiMo-VL-7B-RL-2508 | 38.60 | 84.46 | 65.72 | 56.17 | 65.16 |
| Qwen3-VL-8B-Thinking | 41.20 | 86.42 | 67.20 | 59.54 | 66.75 |
| GLM-4.6V | 42.24 | 85.74 | 69.40 | 58.94 | 68.75 |
| GLM-4.6V-Flash | 45.09 | 86.87 | 70.55 | 61.95 | 69.95 |
| Qwen3-VL-30B-A3B-Thinking | 45.35 | 88.02 | 70.47 | 62.72 | 69.70 |
| Qwen3-VL-235B-A22B-Instruct | 46.26 | 87.87 | 71.07 | 63.38 | 70.59 |
| Step3-VL-10B | 46.83 | 89.01 | 69.69 | 65.03 | 69.09 |
| Qwen3-VL-235B-A22B-Thinking | 48.59 | 88.98 | 72.69 | 65.57 | 72.06 |
| Kimi-K2.5 Thinking | 53.46 | 90.45 | 75.55 | 69.22 | 75.07 |
| Qwen3.5-397B-A17B | 61.95 | 91.67 | 79.31 | 75.81 | 79.01 |
| *Proprietary Models* | | | | | |
| GPT-4o-2024-08-06 | 12.88 | 66.14 | 43.96 | 29.71 | 43.68 |
| Seed-1.5-VL | 42.61 | 86.04 | 70.29 | 58.77 | 69.65 |
| Seed-1.6 | 44.71 | 86.12 | 69.68 | 61.27 | 69.15 |
| Gemini-2.5-Flash | 53.57 | 90.80 | 74.82 | 70.40 | 74.39 |
| Seed-1.8 | 56.84 | 91.09 | 76.70 | 72.14 | 76.30 |
| GPT-5.2 Thinking | 57.36 | 91.38 | 77.25 | 73.18 | 76.10 |
| Gemini-2.5-Pro | 57.98 | 92.41 | 76.65 | 75.63 | 76.19 |
| Seed-2.0-Pro | 63.58 | 93.04 | 81.73 | 76.40 | 81.34 |
| Gemini-3-Pro | 66.90 | 94.48 | 82.64 | 80.15 | 82.21 |

complex logical reasoning. These rubrics are specifically designed to be "Easy-Wrong" for current large multi-modal models, effectively exposing hallucinations and subtle comprehension errors. For instance, in the "Structured Data" and "STEM & Expert" cases, the rubrics require precise reading of axis scales, curve styles, and hierarchical biological relationships, which demand a high level of visual-logical alignment.

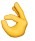

**Natural Scene**

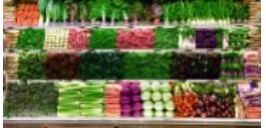

- **a display of vegetables or fresh produce**
- **the items are arranged on shelves, racks, or in a multi-tiered display**
- **specific types of vegetables visible in the image, such as leafy greens, cabbage, broccoli, carrots, or celery**

- a figure **pointing or gesturing** with **an extended arm** (pointing left)
- A figure **kneeling or crouching** at the bottom right, specifically mentioning **an orange, rust, or red skirt**
- a rider wearing **a blue garment** (coat or cloak)
- A figure wearing **a red cap or head covering**
- **a black dog** in the foreground
- must **NOT** mention the presence of **text, signage, or modern vehicles**
- the display consists of **four horizontal shelves or tiers**
- the vegetable in the bottom-left corner as **broccoli**
- must **NOT** identify the vegetable in the bottom-left corner as **Brussels sprouts**
- the block of long, pale green stalks arranged horizontally on the

- bottom shelf as **celery**
- round **purple cabbages** on the bottom shelf
- round **pale green (or white) cabbages** on the bottom shelf
- **red root vegetables** (e.g., radishes) on the second shelf from the top
- **white or pale round bulbs** (e.g., turnips, onions) on the second shelf from the top
- must **NOT** describe the **round bulbs** on the second shelf as **yellow**
- the vertical stacks of bagged **orange vegetables** (carrots) located on the far left and/or far right edges of the display
- the **wooden object** (e.g., chair back, crate, cart handle) visible in the bottom center foreground
- must **NOT** mention the presence of **people** in the image

**Document & OCR**

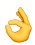

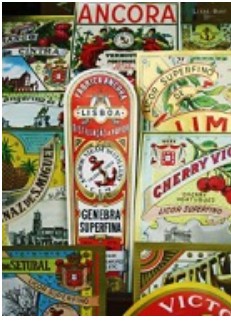

- the core subject of the image as **a collage, collection, or assortment of labels** (specifically recognizing them as liquor, beverage, or vintage labels)
- the prominent **text 'GENEBRA SUPERFINA'** located on the central vertical label
- the large text **'ANCORA'** appearing at the top of the composition
- **fruit illustrations**, identifying at least one specific fruit such as pineapples, cherries, or strawberries

- the volume indicated on the central label as **"750 ml"**
- the fruit illustrated on the label directly to the left of the central label as **a pineapple**
- the fruit illustrated on the label to the right (with text "CHERRY VIG") as **cherries**
- **two women** (or figures) on the top-center "ANCORA" label
- must **NOT** describe the background of the top-center "ANCORA" label as **green** (it is white)
- bare **tree branches** on the bottom-left label
- the watermark text in the upper-right corner as **"LIEBE GABY 2011"**

**Digital UI & UX**

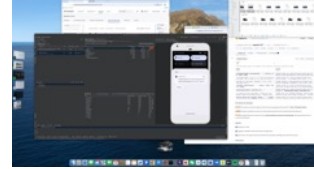

- the operating system environment as **macOS** or mentions the presence of **the Dock** at the bottom
- the central, dark-themed application window as **Android Studio**
- **a mobile device emulator or phone interface** overlaying the main window
- the background **browser window** displaying 'Hugging Face' or 'Open-SFT'
- the mobile emulator is displaying **the notification shade** or mentions visible **toggles** like 'Internet' and 'Bluetooth'

- must **NOT** identify the application environment as **Xcode or an iOS development setup**
- the mobile emulator overlay as **an Android device**
- **the notification text** on the emulator screen as referring to 'Configure physical keyboard' or containing the word 'keyboard'
- the central pane of the IDE as a **'Device File Explorer'** or **a file list** displaying columns such as 'Name', 'Permissions', 'Date', or 'Size'
- the number **'28'** displayed on the Calendar icon in the bottom dock

**Structured Data**

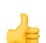

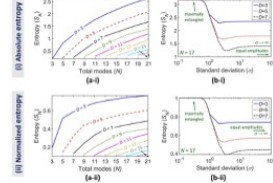

- the x-axis variable for the left-column plots as **'Total modes'** (or N) and for the right-column plots as **'Standard deviation'** (or σ)
- the row labels (or y-axis contexts) as **'Absolute entropy'** and **'Normalized entropy'**
- the curves in the left-column plots show **an increasing trend** as N increases
- the curves in the right-column plots show **a decreasing trend** (or a drop from a high value to a lower plateau) as the standard deviation increases
- the different colored/styled curves represent **different values of 'D'** (e.g., D=3, D=5)

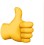

- the x-axis label for the right-column plots as **'Standard deviation'**
- the x-axis in the right-column plots uses **a logarithmic scale**
- the curve corresponding to D=7 in the left-column plots as **a solid black line** (NOT red or dashed)

- the curve corresponding to D=5 in the right-column plots as **a red dotted line** (NOT dashed)
- the green text annotation **'maximally entangled'**
- the green text annotation **'N = 17'** in the right-column plots

*Figure 12.* Qualitative examples of the fine-grained rubrics across four categories: Natural Scene, Document & OCR, Digital UI & UX, and Structured Data. Each example consists of an image and two tiers of rubrics: *Must-Right* (top group) focusing on core facts, and *Easy-Wrong* (bottom group) focusing on challenging details, negative constraints, and logical reasoning.

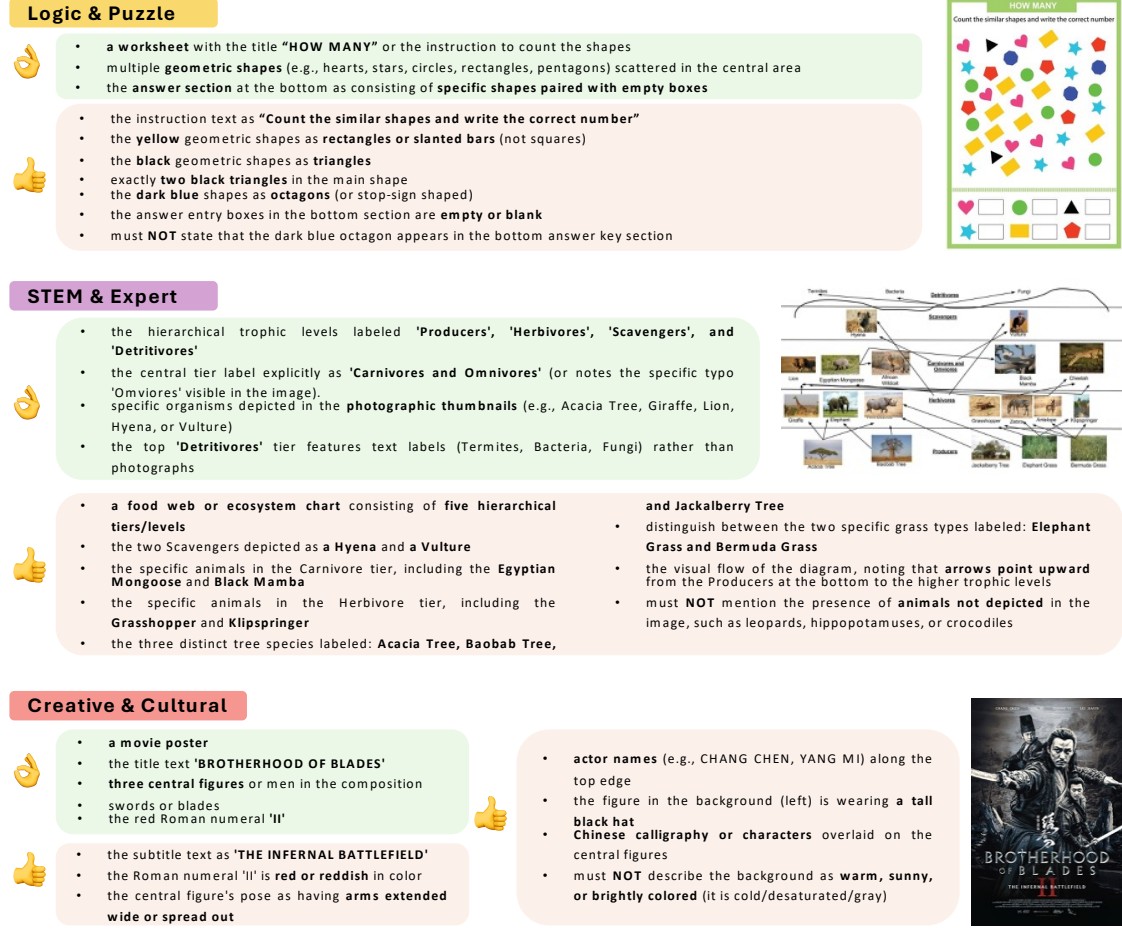

**Logic & Puzzle**

👌
- **a worksheet** with the title **"HOW MANY"** or the instruction to count the shapes
- multiple **geometric shapes** (e.g., hearts, stars, circles, rectangles, pentagons) scattered in the central area
- the **answer section** at the bottom as consisting of **specific shapes paired with empty boxes**

👍
- the instruction text as **"Count the similar shapes and write the correct number"**
- the **yellow** geometric shapes as **rectangles or slanted bars** (not squares)
- the **black** geometric shapes as **triangles**
- exactly **two black triangles** in the main shape
- the **dark blue** shapes as **octagons** (or stop-sign shaped)
- the answer entry boxes in the bottom section are **empty or blank**
- must **NOT** state that the dark blue octagon appears in the bottom answer key section

**STEM & Expert**

👌
- the hierarchical trophic levels labeled **'Producers', 'Herbivores', 'Scavengers', and 'Detritivores'**
- the central tier label explicitly as **'Carnivores and Omnivores'** (or notes the specific typo 'Omviores' visible in the image).
- specific organisms depicted in the **photographic thumbnails** (e.g., Acacia Tree, Giraffe, Lion, Hyena, or Vulture)
- the top **'Detritivores'** tier features text labels (Termites, Bacteria, Fungi) rather than photographs

👍
- **a food web or ecosystem chart** consisting of **five hierarchical tiers/levels**
- the two Scavengers depicted as **a Hyena** and **a Vulture**
- the specific animals in the Carnivore tier, including the **Egyptian Mongoose** and **Black Mamba**
- the specific animals in the Herbivore tier, including the **Grasshopper** and **Klipspringer**
- the three distinct tree species labeled: **Acacia Tree, Baobab Tree, and Jackalberry Tree**
- distinguish between the two specific grass types labeled: **Elephant Grass and Bermuda Grass**
- the visual flow of the diagram, noting that **arrows point upward** from the Producers at the bottom to the higher trophic levels
- must **NOT** mention the presence of **animals not depicted** in the image, such as leopards, hippopotamuses, or crocodiles

**Creative & Cultural**

👌
- **a movie poster**
- the title text **'BROTHERHOOD OF BLADES'**
- **three central figures** or men in the composition
- swords or blades
- the red Roman numeral **'II'**

👍
- the subtitle text as **'THE INFERNAL BATTLEFIELD'**
- the Roman numeral 'II' is **red or reddish** in color
- the central figure's pose as having **arms extended wide or spread out**

👍
- **actor names** (e.g., CHANG CHEN, YANG MI) along the top edge
- the figure in the background (left) is wearing **a tall black hat**
- **Chinese calligraphy or characters** overlaid on the central figures
- must **NOT** describe the background as **warm, sunny, or brightly colored** (it is cold/desaturated/gray)

*Figure 13.* Qualitative examples of the fine-grained rubrics across three additional categories: Logic & Puzzle, STEM & Expert, and Creative & Cultural. Each example consists of an image and two tiers of rubrics: *Must-Right* (top group) focusing on core facts, and *Easy-Wrong* (bottom group) focusing on challenging details, negative constraints, and logical reasoning.

