# OpenReview forum: "PerceptionRubrics: Calibrating Multimodal Evaluation to Human Perception"
_ICML.cc/2026/Conference — ICML 2026 regular_

### Official Review · Reviewer_Keee · 2026-03-05

**Soundness:** 3
**Presentation:** 3
**Significance:** 3
**Originality:** 2
**Overall Recommendation:** 4
**Confidence:** 4

**Summary:**

This paper introduces **PerceptionRubrics**, a rubric-based benchmark for evaluating the perceptual reliability of multimodal large language models (MLLMs). The framework decomposes image understanding into **atomic perceptual statements** derived from captions and evaluates model outputs using structured rubrics with a gated scoring mechanism (“Must-Right”). The authors construct a benchmark spanning seven perception domains and evaluate multiple MLLMs. Results reveal a **reliability gap** between atomic correctness and overall success, suggesting that models often capture individual details but fail to maintain globally consistent perception.

**Compliance With Llm Reviewing Policy:**

Affirmed.

**Final Justification:**

The paper addresses a genuine gap in MLLM perception evaluation with a well-engineered rubric-based pipeline. My main concerns regarding implementation details, potential circularity, and annotation reliability have been substantially addressed in the rebuttal—particularly the detailed rubric post-processing workflow, role-separation design, and the follow-up IAA analysis. While I still consider the methodological novelty somewhat incremental—the contribution is primarily a carefully engineered benchmark rather than a fundamentally new technique—the rigor of construction, strong human alignment, and practical value to the community outweigh this limitation. The rebuttal positively changed my assessment, and I support acceptance.

**Key Questions For Authors:**

**Q1. Rubric generation reproducibility**

The paper describes a caption-centric pipeline for generating rubrics, including the use of rubric proposers and a circular peer-review process to refine captions. While the high-level procedure is outlined, it would be helpful to provide additional implementation details to improve reproducibility. For example, are the rubric generation prompts and filtering rules fixed across domains, and are there post-processing steps (e.g., deduplication or conflict resolution) applied to the generated rubrics?

---

**Q2. Annotation verification and reliability**

The paper mentions the use of advanced reasoning models together with human expert verification to ensure rubric quality. Could the authors provide more details on the verification process? In particular, it would be useful to know whether multiple annotators were involved, whether any inter-annotator agreement (IAA) was measured, and what proportion of rubrics were manually verified.

---

**Q3. Generalization beyond captioning tasks**

The current evaluation framework focuses on caption generation outputs. However, multimodal perception is also critical in tasks such as visual question answering, visual reasoning, and grounding. How do the authors envision extending the rubric-based evaluation framework to these settings?

---

**Q4. Sensitivity of the scoring mechanism**

The paper introduces the Must-Right gating mechanism and shows that it correlates better with human rankings than atomic accuracy. Could the authors further analyze the sensitivity of results to the choice of gating rules? For instance, would alternative scoring mechanisms (e.g., soft gating or weighted constraints) lead to different model rankings or reliability conclusions?

**Limitations:**

The paper would benefit from a clearer discussion of the potential limitations and risks of the proposed benchmark. In particular, the authors could consider:

* Discussing possible bias introduced by caption-based rubric generation, and how caption quality might affect rubric correctness.
* Providing more details on the reliability of rubric annotations (e.g., human verification procedures or agreement statistics).
* Discussing the scalability of the rubric construction pipeline if the benchmark is expanded to larger datasets.
* Clarifying whether the evaluation framework can generalize beyond captioning-based evaluation to other multimodal tasks (e.g., VQA or visual reasoning).

**Strengths And Weaknesses:**

## Strengths

**1. Clear motivation: perceptual reliability is under-measured**

The paper identifies an important limitation of current multimodal benchmarks: high benchmark scores do not necessarily imply reliable perception. Existing benchmarks often rely on multiple-choice formats or similarity-based metrics, which can allow models to exploit linguistic priors or dataset biases rather than demonstrate genuine visual grounding. The paper provides a clear motivation for developing evaluation frameworks that more directly test perceptual consistency.

**2. Rubric-based decomposition enables fine-grained perceptual evaluation**

The proposed rubric-based evaluation framework decomposes visual understanding into atomic, verifiable perceptual statements. This design enables fine-grained diagnosis of perceptual errors, such as hallucinated objects or incorrect attributes, which are difficult to capture using standard benchmark metrics. The approach is conceptually meaningful and aligns with emerging directions in rubric-based evaluation and structured reward modeling.

**3. Insightful analysis of the “reliability gap”**

The paper introduces and empirically demonstrates a “reliability gap” between high atomic accuracy and low strict success rates under the Must-Right gating mechanism. This finding suggests that current multimodal models may correctly identify individual details but still fail to maintain globally consistent perception. The analysis provides a useful diagnostic perspective on the limitations of current MLLMs.

**4. Human alignment validation**

The benchmark evaluation includes human ranking experiments that compare rubric-based scores with human judgments. Demonstrating alignment with human evaluation is an important component for validating the usefulness of a new evaluation framework.

---

## Weaknesses

**1. Limited methodological novelty**

While the rubric-based evaluation framework is applied to multimodal perception, the core ideas largely build upon existing directions such as rubric-based evaluation, structured scoring, and LLM-based assessment. The main contribution appears to lie in benchmark construction and evaluation design rather than introducing fundamentally new evaluation theory or methodology.

**2. Limited reproducibility of the rubric generation pipeline**

The paper outlines a caption-centric pipeline involving rubric proposers and a circular peer-review process for generating rubrics. However, several implementation details necessary for full reproducibility remain unclear, such as the filtering or deduplication rules for generated rubrics, how conflicts between candidate rubrics are resolved, and whether prompts and generation settings are fixed across domains.

**3. Potential evaluation circularity**

The evaluation framework relies on a pipeline in which captions are used to generate rubrics, and model-generated captions are then evaluated against these rubrics. If captions used for rubric construction contain biases or incomplete coverage of visual information, this may introduce evaluation circularity and potentially bias the scoring process. The paper does not sufficiently analyze this risk.

**4. Limited task coverage**

The current benchmark focuses primarily on caption generation outputs. However, perceptual reliability is also critical in other multimodal tasks such as visual question answering, visual reasoning, and grounding. It remains unclear whether the rubric-based evaluation paradigm would generalize effectively to these settings.

---

> ### Author Rebuttal · Authors · 2026-03-31
>
> Thank you for the thoughtful feedback. We appreciate the reviewer’s recognition of our **clear motivation, valuable rubric-based evaluation, insightful analysis, and human alignment validation**. Below, we carefully address each concern.
>
> ``🔗 Tab & Fig``: https://perceptionrubrics.netlify.app/
> # Paper Novelty (W1)
> We agree that our novelty centers on a new benchmark construction method and evaluation design for MLLM perception:
> 1. PerceptionRubrics **addresses an unresolved problem** in multimodal perception evaluation: existing benchmarks are saturated and their rankings do not faithfully reflect global perceptual reliability.
> 2. We introduce **a new perception evaluation pipeline** combining (1) high-information-density data curation, (2) multi-model cross-review for rubric generation, and (3) gated scoring calibrated to critical perceptual failures, turning coarse holistic scoring into fine-grained, human-calibrated rubric auditing.
> 3. The benchmark **yields insightful analysis**, including strong human alignment, the reliability gap, and robustness to superficial biases such as length.
> # More Inplementation Details (W2,Q1,Q2)
> We thank reviewer and will add more details in the revision:
> 1. **Domain-Specific Prompting**: Different visual domains use different prompts for rubric generation and filtering.(line 267-271)
> 2. **Rubric post-processing**: After generation, rubrics go through a post-processing pipeline.\
> (1) Generated rubrics are checked by GPT-OSS-120B to remove semantic duplicates (retain the rubric that is more specific or information-rich).\
> (2) Lightweight human refinement are then conducted to:
>    - Remove duplicate, vague, or non-visual rubrics
>    - Edit weakly grounded rubrics
>    - Verify and resolve conflicts between rubrics, discarding inappropriate ones
>    - Check whether each rubric is correctly categorized (MR/EW), and revise or remove it accordingly
>    - Re-check and add missing rubrics if necessary
> 3.  **Rubric quality verification details**: We hired 9 experts in MLLM research to perform the human refinement described above. Each annotator reviewed approximately 155 samples (while some were discarded finally). Because quality control involves a complex combination of removal, editing, and addition, the process is implemented as structured single-annotator verification per sample. On average, for each sample, annotators removed 2.0, edited 2.3, and added 1.5 rubrics. See Re-Fig 4 for user interface.
>
> # Avoiding Circularity (W3)
> PerceptionRubrics strictly avoid circularity through: (1) Role separation across caption construction, rubric generation, and final evaluation, so that  **no single model family is allowed to both define and score itself**; and (2) **stage-wise de-biasing / human validation** throughout the construction pipeline.
>
> Concretely, for (2):
> 1. **Caption Construction**
>  - Golden captions are produced through heterogeneous circular peer-review across frontier MLLMs, followed by unanimous-consensus filtering and lightweight human verification.
>  - In a 100-sample human evaluation, captions after circular review achieved a 87% win rate over captions from a single strong model, showing that this step **improves alignment with human preference**. See Re-Fig 2 for the user Interface.
>
> 2. **Rubric Construction**\
> All rubrics are further post-processed through **model-based filtering and human-assisted correction**, as detailed in A2 for W2/Q1/Q2.
>
> 3. **Final Evaluation**
> Scoring is performed over fixed atomic rubrics by **an independent LLM judge**, not by the models used for benchmark construction. Replacing the main judge with an alternative strong open-source judge preserves the same ranking order (Fig.8(c)), showing the ranking is **not biased on single evaluator**.
>
> # Task Selection & Rubric Generalization (W4,Q3)
> 1. **Task Selection**:\
> Dense captioning is a foundational open-ended perception task, where evaluation remains challenging and prior benchmarks are often inadequate (Fig 1).
> 2. **Rubric Generalization**:
>  - Rubric-based evaluation is most valuable for open-ended outputs like long-form visual reasoning. The same idea can extend by defining claim-level or step-level rubrics to test whether reasoning is visually grounded and complete.
>  - For tasks such as VQA, grounding, or closed-ended reasoning, where a canonical answer is usually available, direct task-specific metrics are often sufficient. A rubric layer may be useful as a diagnostic supplement here.
>
> #  Scoring Mechanism Sensitivity (Q4)
> We compared the **soft-weighted scoring** rule that combines Must-Right and Easy-Wrong, and evaluated rankings against VisionArena, a well-recognized human-annotated vision benchmark. In Re-Tab 6, ρ shows the induced model ranking remains stable. At the same time, **the strict gate** yields larger score dispersion, meaning it better separates models, reduces near-tie compression, and **provides stronger discriminative resolution for critical perceptual failures**.

---

> > ### Author Rebuttal · Reviewer_Keee · 2026-04-01
> >
> > I thank the authors for the detailed rebuttal. My main concerns have been substantially addressed.
> >
> > I still believe the methodological novelty is somewhat limited, and the annotation reliability discussion would be even stronger with explicit agreement statistics. However, these concerns are no longer sufficient for me to maintain my original overall assessment.

---

> > > ### Author Response · Authors · 2026-04-08
> > >
> > > Thank you very much for your support and thoughtful follow-up. We sincerely appreciate your acknowledgment that the main concerns have been substantially addressed.
> > >
> > > We also appreciate your suggestion that the reliability would be stronger with explicit agreement statistics. We added a complementary human evaluation for Must-Right rubrics together with inter-annotator agreement (IAA).
> > >
> > > **1. Setting:**
> > >
> > > We randomly sampled 100 cases and asked 3 professional annotators to independently judge, for each case, whether each current Must-Right rubric is both correct and necessary for the image. The final decision for each rubric was determined by majority vote.
> > >
> > > **2. Results and IAA:**
> > >
> > > Across the total 396 annotated Must-Right rubrics, **98.99%** were confirmed by majority vote. In **96%** of the sampled cases, all Must-Right rubrics passed the human review. We compute **IAA** on these rubric-level judgments, obtaining ***Unanimous agreement rate** = **97.47%***, which indicates strong agreement among annotators.
> > > These high agreement and pass-rate demonstrate the reliability of Must-Right rubrics.
> > >
> > > Thank you again for your careful reading and raising your score.

---

### Official Review · Reviewer_LJov · 2026-03-09

**Soundness:** 3
**Presentation:** 2
**Significance:** 3
**Originality:** 3
**Overall Recommendation:** 4
**Confidence:** 3

**Summary:**

This study focuses on a central concept: improving the evaluation of MLLMs by aligning benchmark metrics with human perceptual judgment. Current perception benchmarks often produce saturated leaderboards while models still fail in fine-grained visual reasoning. To address this gap, the authors introduce PerceptionRubrics, a rubric-based evaluation framework built from 1038 high-informaiton images and more than 13000 atomic evaluation rubrics. These rubrics are derived from "golden captions" produced through a circular peer-review process involving multiple MLLMs and human verification. The evaluation metric introduces a gated scoring mechanism where essential perceptual facts act as a strict gate: failing any essential fact results in a zero score, while secondary rubrics (Easy-Wrong) differentiate among passing outputs. Experiments across 19 models reveal a "reliability gap" between atomic recognition and holistic correctness, and demonstrate stronger alignment with human judgment compared to existing benchmarks. Overall, the work proposes a new paradigm for multimodal evaluation based on atomic perception auditing rather than holistic semantic similarity.

**Compliance With Llm Reviewing Policy:**

Affirmed.

**Final Justification:**

Thank you for the authors’ response.  Although the authors have added human evaluation, there is still a lack of rigorous theoretical proof. The authors may consider whether there are better approaches. Overall, I will maintain my original score. Thanks again.

**Key Questions For Authors:**

It is unclear whether the authors have measured the time required for the full evaluation process. Since the framework relies on multiple rubrics to guide the evaluation, the procedure may be relatively time-consuming. Reporting the evaluation or inference time for a single model would help clarify the practical efficiency of the proposed framework.

**Limitations:**

The paper discusses the potential negative societal impact of the work but does not provide its limitations. Providing a discussion of the limitations would help readers better understand the scope and potential constraints of the proposed approach.

**Strengths And Weaknesses:**

### Strengths

1. The paper addresses a real and growing problem in multimodal research. Current benchmarks rely heavily on holistic similarity metrics and fail to penalize localized hallucinations. The paper correctly identifies this as an evaluation misalignment with human perception.

2. The core idea is decomposing perception into atomic visual facts: Must-Right and Easy-Wrong. This approach resembles unit testing for perception. This decomposition improves evaluation granularity compared to typical caption metrics.

3. The experimental analysis reveals several interesting findings. This findings is valuable for future research.

### Weaknesses

1. It is unclear how the reliability of the Must-Right rubrics is ensured. Since the evaluation metric assigns a zero score if any Must-Right criterion is violated, inaccuracies in these rubrics could significantly affect the evaluation results.

2. In Figure 7, the benchmark score increases by nearly 30% from GPT-4o to Qwen3-VL-8B, while the Human Expert score only rises slightly from 93% to 97%. Although the overall trend is consistent, the magnitude of the difference between the two is noticeably large.

3. The dataset construction relies heavily on MLLMs. Even though humans verify final captions, the pipeline is still largely model-driven.

4. The paper's formatting could be improved, as the text and the corresponding figures/tables are often placed far apart, which makes the content harder to follow.

---

> ### Author Rebuttal · Authors · 2026-03-31
>
> Thank you for the constructive feedback. We appreciate the reviewer’s **recognition of the importance of the problem, the value of our rubric-based evaluation, and the usefulness of our experimental findings for future research**. Below, we carefully address each concern and clarify potential misunderstandings.
>
> ``Tab & Fig``: https://perceptionrubrics.netlify.app/
> # Rubrics Reliability (W1)
> After generation, rubrics go through a **post-processing pipeline to ensure reliability** of both the must-right and easy-wrong rubrics.
> 1. Model-assisted filtering:\
> Generated rubrics are first checked by GPT-OSS-120B to remove semantic duplicates, retaining the more specific and information-rich rubric.
> 2. Lightweight human refinement are then conducted to:
>    - Remove duplicate, vague, or non-visual rubrics
>    - Edit weakly grounded rubrics
>    - Verify and resolve conflicts between rubrics, discarding inappropriate ones
>    - Check whether each rubric is correctly categorized (Must-right / Easy-wrong), and revise or remove it accordingly
>    - Re-check and add missing rubrics if necessary
>
> **Rubric quality verification details**: \
> We hired 9 experts in MLLM/LLM research to perform the human refinement described above. The user Interface is shown in Re-Fig 4. Each annotator reviewed approximately 155 samples (while some were discarded finally). On average, for each sample, annotators removed 2.0, edited 2.3, and added 1.5 rubrics, ensuring the final rubric set is human-calibrated.
>
> # Magnitude Discrepancy (W2)
> We sincerely appreciate this observation. The two axes in Fig. 7 are on different scales. The Human Expert score is derived from an Elo-style pairwise ranking protocol, which is primarily **meaningful for relative ranking rather than absolute score magnitude**. Specifically, Elo ratings are designed solely to reflect the probability of winning in head-to-head matchups within a specific pool of competitors; thus, the numerical gaps are non-linear, zero-sum, and lack an absolute baseline for capability. In contrast, our benchmark score is an absolute metric grounded in rubrics.
>
> Therefore, the two are **not expected to match numerically**. The primary purpose of Fig 7 is to test whether the benchmark successfully induces the **same ordering and capability trend as human judgment**.
>
> # On the Model-Driven Construction (W3)
> We agree that the benchmark construction is model-assisted, but we would like to clarify that it is **not model-determined**. Our goal is to build an effective automated benchmark construction pipeline that **reduces construction cost while keeping humans in the loop for calibration and quality control** (see Re-Fig. 1 and Re-Fig. 2 for the comparison between model-based and fully manual construction costs).
>
> To ensure construction reliability, PerceptionRubrics is explicitly designed around human-calibrated role separation:(1) Role separation across caption construction, rubric generation, and final evaluation, so that **no single model family is allowed to both define and score itself**; and (2) **stage-wise de-biasing / human validation** throughout the construction pipeline. Concretely, for  (2):\
> **1. Caption Construction**
>  - Golden captions are produced through heterogeneous circular peer-review across frontier MLLMs, followed by unanimous-consensus filtering and lightweight human verification.
>  - In a 100-sample human evaluation, captions after circular review achieved a 87% win rate over captions from a single strong model, showing that this step **improves alignment with human preference**. See Re-Fig 2 for the user Interface.
>
> **2. Rubric Construction**
>   - All rubrics are further post-processed through model-based filtering and manual correction to ensure **correctness and human alignment**, as detailed in the answer for W1.
>
> **3. Final Evaluation**
>  - Scoring is performed over fixed atomic rubrics by an independent LLM judge, not by the models used for benchmark construction. Replacing the main judge with an alternative strong open-source judge preserves the same ranking order (Fig.8(c)), showing the ranking is **not biased on single evaluator**.
>
>
> # Paper Formating (W4)
> Thank you for the suggestion. We will improve the formatting in the revision by placing text closer to the corresponding figures/tables and improving readability.
>
> # Evaluation Time (Q1)
> To quantify practical efficiency, we measured the full evaluation time for Qwen3-VL-8B. The judge model, GPT-OSS-120B, was deployed on 2 nodes with 4 H100 GPUs each. Under 64-way concurrency, judging the full benchmark takes 524s in total, corresponding to about 0.5s per sample, which confirms that **the evaluation pipeline is efficient in practice**.

---

> > ### Author Rebuttal · Reviewer_LJov · 2026-04-03
> >
> > Thank you for the authors’ response. Although the authors reiterate the reliability of Must-Right in their rebuttal, there is still insufficient evidence to convincingly support its reliability. Therefore, I maintain my original rating.

---

> > > ### Author Response · Authors · 2026-04-08
> > >
> > > Thank you for your support and patient further explanation. We fully understand your concern that the reliability of Must-Right rubrics requires **more direct evidence**, and we therefore add a complementary human evaluation specifically for this point.
> > >
> > > **1. Must-Right reliability human evaluation.**
> > >
> > > We randomly sampled 100 cases and asked 3 professional annotators to independently judge, for each case, whether each current Must-Right rubric is correct and necessary for that image. The final decision for each rubric was determined by **majority vote**.
> > >
> > > **2. Results.**\
> > > Across the total 396 annotated Must-Right rubrics, **98.99%** were confirmed by majority vote. In **96%** of the cases, all Must-Right rubrics passed the human review.
> > > These results provide direct supporting evidence that the Must-Right rubrics owns **high-reliability** for indispensable perceptual facts.
> > >
> > > Thank you again for this helpful suggestion. We will further improve the benchmark quality and make this validation more explicit in the revision.

---

### Official Review · Reviewer_pVFd · 2026-03-11

**Soundness:** 3
**Presentation:** 4
**Significance:** 2
**Originality:** 2
**Overall Recommendation:** 3
**Confidence:** 4

**Summary:**

The paper proposes PerceptionRubrics, a rubrics-based evaluation framework that applies a gated scoring mechanism for benchmarking “must-right” and “easy-wrong” aspects in captions generated by a VLM. The paper performs experiments along a wide-variety of VLMs, and finds that while the current model verifies isolated elements correctly they fail conjunctive constraints, proprietary closed source models do generally better, certain domains are harder than others.

**Compliance With Llm Reviewing Policy:**

Affirmed.

**Final Justification:**

I appreciate the authors’ efforts during the rebuttal in providing additional clarifications and resolving some of the concerns. As a result, I have slightly increased my score. However, after considering the rebuttal and the discussion, including other reviewers’ comments, I overall remain unconvinced that the paper offers sufficient methodological novelty or new insights to meet the acceptance bar.

**Key Questions For Authors:**

- How does the proposed gated scoring differ in practice from prior hallucination benchmarks that already measure correctness and hallucination via precision/recall style metrics? Examples where your rubric changes model rankings would clarify the novelty.

- Why is a strict zero score used if any criterion fails? Did you explore weighted or hierarchical rubrics where different facts have different importance, and how would that affect human alignment?

- Could you provide more details about the human verification step in the circular peer review pipeline (who annotated, annotation effort, and fraction of captions requiring edits)? This is important for understanding dataset cost and scalability.

- Since captions are generated using GPT-5.2, Gemini 3-Pro, and Seed-1.8, is there a risk the benchmark favors these model families? Have you tested strong models outside this set to check for potential bias?

- The human alignment analysis focuses on model ranking correlation. Could you also report sample level agreement with human scores and inter annotator agreement to better understand alignment and subjectivity?

**Limitations:**

I did not see a substantial discussion of limitations or future work in the paper. It would be helpful for the authors to expand on this, particularly given the reliance on an MLLM as judge setup, which can introduce its own evaluation errors, and the heavy use of proprietary models, which may make it expensive to scale the benchmark or iterate on it in the future.

**Strengths And Weaknesses:**

Strengths:
- The paper is well written and easy to follow.
- The evaluation includes a broad set of models, enabling useful insights such as differences between open and closed source systems and performance variation across categories.

Weaknesses:

- The paper motivates the rubric design by arguing that existing benchmarks rely on linear scoring mechanisms that dilute meaningful differences between models. However, it is not clear that the proposed gated scoring is substantially different from prior hallucination benchmarks such as [1,2], which already evaluate correctness of ground truth facts and hallucination propensity separately. These benchmarks effectively penalize hallucinations through precision, recall, and F1 style metrics, so it is unclear whether the proposed formulation provides fundamentally new evaluation behavior.

- Additionally, the strict gating rule (score zero if any criterion fails) may be overly rigid. In practice, not all factual elements in a caption are equally important. If the goal is more atomic evaluation, I believe it may be more informative to weight or prioritize different facts rather than treating all criteria as equally critical

- Claims about limitations of existing benchmarks: The paper states that captions containing hallucinations can still receive high scores under existing metrics. This claim would benefit from stronger justification. Many hallucination oriented benchmarks explicitly measure hallucination precision and recall, and their combined metrics already penalize hallucinated content.

- Potential benchmark bias. The same model families used to generate captions also perform strongly on the benchmark. This raises the possibility that the dataset may favor similar architectures or training distributions. It would be useful to test strong models from other families (for example Claude class models) to check whether the benchmark introduces unintended bias.

- Human verification details. The paper mentions a human correction loop but provides limited information about it. For example:
   - Who performed the verification (authors vs. crowdworkers)?
   - How much time or effort was required?
   - What proportion of captions required no edits, minor edits, or major corrections?

- Also, It is not clear how much benefit the circular review process provides compared to simply using a single strong model with human verification. I think that quantitative comparisons or ablations here are important to understand how useful are key contributions of the paper compared to standard data generating approaches..


- The human alignment analysis focuses primarily on correlation between benchmark based model rankings and human rankings. While useful, this is a relatively coarse metric. For an evaluation benchmark, it would also be helpful to report sample level alignment, such as:
  - Agreement between human scores and benchmark scores for individual captions.
  - Inter annotator agreement among humans at caption level.

Since image captioning is inherently subjective and the images appear perceptually rich, understanding this variance is important to determine whether models are being over penalized or whether disagreements are driven by annotation variability.

- The paper compares human alignment primarily against DOCCI and CAPTURE, both released in 2024. It would be helpful to also compare against more recent captioning benchmarks such as VDC [3] to better position the contribution relative to the current landscape.


[1]  HallusionBench: An Advanced Diagnostic Suite for Entangled Language Hallucination and Visual Illusion in Large Vision-Language Models

[2] ARGUS: Hallucination and Omission Evaluation in Video-LLMs

[3] AuroraCap: Efficient, Performant Video Detailed Captioning and a New Benchmark

---

> ### Author Rebuttal · Authors · 2026-03-31
>
> Thank you for the thoughtful feedback and recognizing our paper’s **clear presentation, comprehensive evaluation, and useful insights**. Below we address each concern and clarify misunderstandings.
>
> ``Tab & Fig``: https://perceptionrubrics.netlify.app/
> # Difference from Mentioned Metrics (W1,W3,Q1)
> We clarify differences from three aspects:
> 1. **Evaluation Focus**\
> PRB targets **case-level global perception** rather than another local hallucination benchmark. By contrast, HallusionBench is a diagnostic VQA-style benchmark focused on **question-pair accuracy and consistency**, while ARGUS diagnoses hallucination and omission for **video** captioning, with emphasis on **temporal dynamics**.
> 2. **Metric Behavior**\
> PRB does **not just separate "correctness" and "hallucination"**; it distinguishes MR and EW by their importance for global perception. In HallusionBench- or ARGUS-style evaluation, local items are aggregated more uniformly, so one critical failure can be **diluted** by correct details. In contrast, PRB uses a **non-compensatory** gate: one MR failure cannot be averaged away by correct EW facts (Lines 77–82).
> 3. **Ranking Behavior**\
> For **comparison with an F1-style metric**, the main paper already shows that CAPTURE produces compressed scores even for models with large capability gaps (Fig 1), whereas PRB **better separates** such models and also shows substantially **stronger human alignment** (Fig 6).
> # On Gated Scoring (W2,Q2)
> 1. We **clarify “Not all facts are equally important” is exactly the design principle** of PRB: Must-Right captures foundational, non-negotiable facts for case-level acceptability, while Easy-Wrong provides finer-grained differentiation. The strict gate applies only to MR, not to all facts.
> 2. Following reviewer’s suggestion, we tested the **soft-weighted scoring** rule and evaluated rankings against VisionArena, a well-recognized human-annotated vision benchmark. As shown in Re-Tab 6, the induced ranking remains highly stable (ρ = 0.983), while the strict gate **better separates models, and reduces near-tie compression**. Thus, strict gating remains the more discriminative choice for critical perceptual failures.
> # Debias Verification (W4,Q4)
> 1. **PRB is structurally de-biased**\
> The pipeline is explicitly decoupled into caption construction, rubric generation, and LLM-as-a-judge, with **dedicated bias-control and human-alignment checks at each stage** (see A3 for Reviewer 2wCg).
> 2. **Family overlap does not inflate benchmark scores**\
> GPT-4o (Tab 1), despite belonging to a construction family, still ranks below models not used in benchmark construction. Moreover, the overall benchmark ranking is highly consistent with **human-annotated ranking** VisionArena (**Pearson / Kendall / pairwise agreement = [0.976, 0.929, 0.964]**), arguing against a construction-family artifact.
> 3. **Cross-Family Sanity Check**\
> We additionally evaluated Qwen3.5-PLUS from another model family, which obtains a strong score of 61.33, further suggesting that the pipeline is **not biased** toward the families used in construction. We will also include Claude-family models in the revision as resources allow.
> # Caption Audit Details (W5,Q3)
> This was performed by 9 experts in MLLM research (Interface: Re-Fig 3). On average, each sample required about 212 seconds for annotation, and the final edits were lightweight local corrections (about 5–6 edit points per caption, usually affecting 1–3 words each), with most edits involving minor number or attribute corrections. This indicates that the process remains **low-cost and practically scalable**.
> # Circular Review Benefits (W6)
> As shown by Re-Fig 2, we compared captions from a single strong model against those from the circular review. In a 100-sample human evaluation, circular-review captions achieved an 87% win rate, showing **improved caption quality and reduced burden of human correction**.
> # Human Alignment Analysis  (W7,Q5)
> We analyzed the sample-level human evaluation on 100 captions from the five models in Fig 7. We recruited 6 experts, split into two independent 3-annotator groups, each evaluating 50 captions using a 5-point scale (Re-Fig 5).
> 1. The mean human scores for models are 2.1, 2.9, 3.3, 3.7, and 4.5, preserving the **same ordering as our benchmark** but with a softer, more compressed scale.
> 2. We further measure **ranking agreement** within each 3-annotator group. The mean pairwise Kendall’s τ-b = 0.80, indicating **strong consistency** in the relative ranking of model quality among humans.
> # Comparison with VDC (W8)
> 1. BMK Focus: For video captioning, VDC emphasizes temporal dynamics, long-form events, and camera motion. PRB targets dense image captioning for global perception under high information density.
> 2. Visual Domain: VDC focuses on open-domain video scenes with natural scenes and human activities, while PRB covers more diverse and information-dense settings, including documents, GUIs, charts, and other structured images.

---

> > ### Author Rebuttal · Reviewer_pVFd · 2026-04-04
> >
> > I thank the authors for the response. However, I am still not fully convinced by the methodological contributions or the insights presented in the paper.
> >
> > My point regarding HallusionBench and ARGUS is that many existing benchmarks already separate critical errors that a model must not get wrong (recall) from less critical ones (precision). The gating mechanism that assigns a score of zero when even a single critical error occurs seems like a straightforward modification that could be applied to many existing evaluation frameworks. In practice, this choice depends on the tolerance of the downstream user and may not be a one size fits all solution.
> >
> > I appreciate the authors' effort to include human grounding through the 100 sample evaluations, and I understand the time constraints during rebuttal. That said, the scale of these evaluations is still quite limited, and they do not yet provide strong confidence in the benchmark. Since the core contribution of the paper is the proposed metric and benchmark, I believe a more rigorous human evaluation would be important, potentially involving a larger sample size and broader participation beyond the authors.

---

> > > ### Author Response · Authors · 2026-04-08
> > >
> > > Thank you for your continued engagement and candid final thoughts. We fully respect your perspective. To ensure clarity for the broader discussion phase, we would like to briefly clarify remaining points: **(1)** the comparison with prior hallucination-oriented benchmarks, **(2)** the benchmark contribution, **(3)** the relationship between the gate design and downstream tolerance, and **(4)** the scale and role of the added human evaluation.
> > >
> > > **1. Clarification on prior benchmark comparison.**
> > > We agree that HallusionBench and ARGUS are relevant prior works, but we would like to clarify that they **do not separate facts by importance/severity in the way PerceptionRubrics (PRB) does**. More specifically, for benchmark differences: :
> > >  - HallusionBench is primarily a **diagnostic VQA-style** benchmark for local inconsistency, using paired questioning to test whether models respond consistently to positive/negative visual evidence. ARGUS measures **hallucination and omission cost** for video captioning at the sentence level. In both cases, the final metric aggregates local errors relatively **uniformly**.
> > >  - By contrast, for PRB:
> > >     - It focuses on the model capability of **global and dense perception**.
> > >     - It explicitly distinguishes Must-Right and Easy-Wrong rubrics according to the **different importance of perceptual facts** under information-rich image understanding, modeling the **nonlinearity** of human perception.
> > >
> > > Hence, PRB is **not equivalent to simply adding a gate** on top of an existing hallucination metric.
> > >
> > > **2. Metric vs. system-level contribution.**\
> > > We agree that the gating formula itself is relatively straightforward, but our **core contribution lies in the system-level benchmark design**:
> > >  - Specifically, the benchmark combines **high-information-density curation, multi-model cross-review for rubric generation, and human-calibrated severity-aware evaluation**.
> > >  - Together, these components **address a practical problem that remains unresolved** in current perception evaluation: existing benchmarks are often saturated, and their rankings do not faithfully reflect global perceptual reliability.
> > >   - It is this full system design, rather than the gate alone, that enables the benchmark to **reveal the reliability gap, achieve strong human alignment, and provide more discriminative model assessment**.
> > >
> > >
> > > **3. One-size-fits-all vs. a strict perception unit test.**\
> > > We agree with the good point that downstream tolerance varies. PRB is therefore not intended as a universal one-size-fits-all scoring rule, but rather as **a strict unit test for foundational perception**. At the same time, because evaluation is decomposed into atomic rubrics, the framework naturally remains flexible: users can apply the strict gate in safety-critical settings, or instead adopt soft weighting over the same pre-compiled rubrics in higher-tolerance settings.
> > >
> > > **4. On human evaluation.**\
> > > We appreciate your point that broader human evaluation would further strengthen the benchmark. At the same time, We would like to emphasize that our **primary human validation** is the benchmark’s **strong ranking alignment** with VisionArena (Pearson = 0.976), which provides large-scale human preference grounding. The 100-sample human studies were added during rebuttal specifically to address the request for intermediate golden-caption evaluation under the available time constraints. We fully agree that larger-scale caption-level human studies would be valuable, and we would be happy to expand this direction in a revision.
> > >
> > >
> > > We truly hope this clarification can be helpful for better understanding. Thank you again for your thoughtful engagement and suggestions.

---

### Official Review · Reviewer_2wCg · 2026-03-12

**Soundness:** 3
**Presentation:** 4
**Significance:** 2
**Originality:** 3
**Overall Recommendation:** 3
**Confidence:** 3

**Summary:**

The paper proposes PerceptionRubrics, a benchmark that better calibrates model judgment toward human evaluation in the image caption task. Concretely, it identifies the gap between current reward signals and human perception, and introduces two rubrics, a must-right one and an easy-wrong one, to mimic human judgment. Through a systematic evaluation of a wide range of MLLMs, the benchmark demonstrates better distinguishability and alignment with humans.

**Compliance With Llm Reviewing Policy:**

Affirmed.

**Final Justification:**

The rebuttal adequately addressed my technical queries regarding evaluation fairness, ranking baselines, and binary separation. However, my core concern regarding cost-effectiveness and contribution remains. In this saturated field, justifying the engineering overhead by comparing it to from-scratch manual annotation is unconvincing, the true baseline should be existing established datasets. Also, the framework provides marginal, incremental benefits that overlap with similar existing benchmarks. While the paper is comprehensive and technically sound, it lacks the standout novelty or significant insights required to broadly inspire future research.

Overall, I maintain my score of 3, though I consider a 4 equally justifiable. I will not strongly champion for or against acceptance.

**Key Questions For Authors:**

Please refer to the weaknesses.

**Limitations:**

I appreciate the elaborately constructed benchmark, yet I think the inflexibility of expanding evaluation should be acknowledged as a limitation (or some other issues, if this is disproved by the rebuttal).

**Strengths And Weaknesses:**

Strength:
1. The paper focuses on the practical issue of performance saturation. The proposed benchmark and metrics offer a targeted and reasonable solution.
2. PerceptionRubrics is demonstrated to be of high quality and diversity.
3. The evaluation on PerceptionRubrics is comprehensive. The results, especially human alignment, make the advantages of the proposed evaluation highly persuasive.

Weakness:
1. My prior concern is the cost-effectiveness of the proposed evaluation strategy. It takes multiple inferences of proprietary MLLMs to obtain the two rubrics. This significantly restricts the flexibility of reproducing the strategy on other image domains. I wonder if this process can be approximated by targeted prompt tuning during the LLM-as-judge time. For instance, directly asking LLMs to identify unignorable mistakes to mimic the must-right rubrics.
2. Another issue is the strong positive correlation between must-right rate and easy-wrong accuracy, as shown in Figure 6. While this sheds light on the inherent connection between MLLMs' two basic properties, this somewhat conflicts with the statement in lines 77-82, as the binary separation aligns with the metric score from a statistical perspective.
3. Some proprietary MLMs are used in both the construction and evaluation phases, which should be clarified to avoid unfairness.
4. (Minor) In my opinion, one of the primary purposes of evaluation metrics is to rank models. From this perspective, can the authors provide some direct comparison between rankings from popular arenas and the proposed metrics? Some corresponding analyses are also desired.

---

> ### Author Rebuttal · Authors · 2026-03-31
>
> Thank you for the thoughtful feedback. We appreciate the reviewer’s recognition of the **reasonable design of our benchmark, its high quality and diversity, and our comprehensive, human-aligned evaluation**. Below, we carefully address each concern and clarify potential misunderstandings.
>
> ``🔗 Tab & Fig``: https://perceptionrubrics.netlify.app/
> # A1. Cost-Effectiveness (W1)
> **1. Cost: Inexpensive and reproducible evaluation strategy**
>
> We'd like to **clarify that this strategy contains two distinct stages**:
> (1) One-time rubric construction
> (2) Recurring evaluation once the rubric set is fixed.
> The costs should therefore be assessed separately:
>  - Inexpensive and fast recurring evaluation
>    - Once rubrics are extracted offline, evaluation reduces to a **lightweight checklist-style procedure**: The judge model, GPT-OSS-120B, was deployed on 2 nodes with 4 H100 GPUs each. Under 64-way concurrency, judging the full benchmark takes 524s in total, corresponding to about 0.5s per sample.
>  - One-time construction is much cheaper than manual annotation
>    - Re-Tab.1 and Re-Tab.2 compares the cost of manual construction against our automated pipeline. Our automated pipeline costs approximately only \$0.11–\$0.15 per image, whereas a fully manual pipeline is estimated at \$15–\$18.34 per image; even a simplified manual process that only writes rubrics still exceeds \$5 per image.
>    - These make the benchmark extensible to new image domains: expansion can be done in a **largely automated way** rather than requiring full manual rubric authoring.
>
> **2. Effectiveness: PerceptionRubrics (PRB) construction vs. targeted prompt tuning**
>
> Following reviewer's suggestions, we implement the mentioned evaluation by designing a **comprehensive prompt** (Re-Fig 1) with GPT-5.2 as a multimodal judger. The results shows:
>  - Lower discriminative power and weaker human alignment. The resulting rankings show substantially lower score separation (Re-Tab. 3) and weaker consistency with VisionArena (Re-Tab. 4), a well-recognized human-annotated vision benchmark, than our PRB. These support the necessity of our offline construction pipeline.
>
>
> # A2. Necessity of Binary Separation (W2)
>
> The Must-Right / Easy-Wrong separation is necessary for our task because it captures **sample-level non-compensatory** acceptability (Lines 77–82), and this is **not in contradiction** with Fig. 6:
> 1. Fig 6 reflects a **model-level** trend: across models, Must-Right pass rate and Easy-Wrong accuracy are highly correlated, meaning stronger models tend to improve in both foundational perception and fine-grained details together.
> 2. Lines 77–82 reflect **sample-level** nonlinearity: for an individual caption, one critical perceptual failure can make the whole prediction unacceptable, even if many other local facts are correct.
>
> In other words, Fig 6 is about co-varying capabilities across models, while our binary separation is about how essential errors should be treated within a single sample.
>
> # A3. Ensuring Fairness (W3)
> To ensure fairness, PRB is explicitly designed around human-calibrated role separation: (1) Role separation across caption construction, rubric generation, and final evaluation, so that **no single model family is allowed to both define and score itself**; and (2) **stage-wise de-biasing / human validation** throughout the construction pipeline to avoid model self-preference. Concretely, for (2):
>
> **1. Caption Construction**
>  - Golden captions are produced through heterogeneous circular peer review, followed by unanimous-consensus filtering and lightweight human verification.
>  - In a 100-sample human evaluation (Re-Fig 2), captions after circular review achieved a 87% win rate over single-model captions, showing that this step **improves alignment with human preference**.
>
> **2. Rubric Construction**
>  - The generated rubrics undergo a two-stage calibration: (1) GPT-OSS-120B removes semantic duplicates while retaining the more specific and information-rich rubric. (2) human experts refine the rubric set by removing duplicate, vague, or non-visual items, editing weakly grounded rubrics, resolving conflicts, correcting Must-Right / Easy-Wrong categorization, and adding missing rubrics when necessary (Re-Fig 4). On average, annotators removed 2.0 rubrics, edited 2.3, and added 1.5 per sample, ensuring the final rubric set is a calibrated.
>
> **3. Final Evaluation**
>  - Scoring is performed over fixed atomic rubrics by **an independent LLM judge**, not by the models used for benchmark construction. Replacing the main judge with an alternative strong open-source judge preserves the same ranking order (Fig.8(c)), showing the ranking is **not biased on single evaluator**.
>
> # A4. Comparison with VisionArena (W4)
> To evaluate ranking quality directly, we compare the rankings from PRB with those from VisionArena. In Re-Fig 5, our benchmark shows the strongest consistency, indicating that it provides a highly human-aligned ranking signal in practice.

---

> > ### Author Rebuttal · Reviewer_2wCg · 2026-04-03
> >
> > I would like to thank the authors for their dedicated rebuttal, which has successfully addressed most of my concerns.
> >
> > However, LLM evaluation, especially the development of metrics for mature tasks, is a highly saturated field. Personally, although the authors have broadly argued for the efficiency of the evaluation phase, I still hold mild reservations regarding the overall flexibility and the engineering effort required. Furthermore, the additional insights provided beyond existing frameworks appear somewhat marginal (primarily the 'must-right' concept), making it difficult for this work to exert a substantial practical impact on well-established evaluation systems.
> >
> > Overall, I consider this a comprehensive paper, yet one that lacks standout contributions. I would strongly lean toward a score of 3.5 if such an option existed, but under the current grading system, my inclination remains a 3.

---

> > > ### Author Response · Authors · 2026-04-08
> > >
> > > We sincerely appreciate that **our response helped fully resolve your concerns**. We also fully understand your remaining considerations, and would like to briefly clarify **(1)** the acceptable cost and necessity of the associated engineering effort, and **(2)** the benchmark contribution in addressing current important evaluation bottlenecks.
> > >
> > > **1. Regarding Engineering Efforts**\
> > > We agree that the one-time benchmark construction requires non-trivial effort, even though the recurring evaluation is efficient. We would like to further clarify its **cost-effectiveness, flexibility, and necessity**:
> > >  - **Lower cost and reasonable flexibility.** The construction pipeline is highly automated, **substantially cheaper** than full manual construction, and reasonably reusable across domains (Re-Tab 1-2). Extending to a new image domain mainly requires **adapting the rubric-generation prompts, rather than redesigning the full pipeline**.
> > >  - **Necessary effort for benchmark quality.** In PerceptionRubrics (PRB), this one-time effort is part of what **enables stronger debiasing, better human alignment, and more discriminative evaluation** than lighter-weight alternatives, as also supported by our comparison between offline PRB construction and targeted prompt-only judging.
> > >
> > > We sincerely appreciate your suggestion and fully agree that lower-cost and more flexible construction would be valuable future directions.
> > >
> > > **2. PRB Contribution**\
> > > We understand that the evaluation landscape may **appear saturated at first glance**. However, PRB identifies an **important yet still unresolved gaps** in this area, and addresses it through a comprehensive benchmark-design methodology. In this sense, the contribution is not a single component such as “Must-Right,” but the **integrated system design, the resulting benchmark behavior, and the empirical insights it enables**.
> > >
> > >  - **Benchmark Focus & Construction.** As discussed in the introduction, dense perception is a foundational VLM capability, yet:
> > >     - Its reliable evaluation across **diverse, information-rich image domains** remains a real **bottleneck**.
> > >     - When existing benchmarks become **saturated**, or **fail to clearly separate weaker from stronger** models, they no longer provide useful guidance for model improvement.
> > >
> > >     In this sense, PRB is meant to **address a concrete evaluation pain point** through **diverse image-task sources, a rubric-based evaluation design, and a highly automated but quality-controlled construction pipeline**—precisely the practical value that we were encouraged to see reflected in your original strengths.
> > >  - **Strong human alignment.** It is especially important to have a perception benchmark with strong human alignment, which has historically been difficult for image caption evaluation. In this respect, PRB’s **strong consistency** with VisionArena, a large-scale and expensive human-annotated benchmark, is particularly valuable: it provides the open community and academia with a more accessible benchmark that better couples research evaluation with practical usefulness.
> > >
> > > This **fills an important gap in the current evaluation ecosystem**.
> > >
> > > We truly hope this clarification can be helpful. Thank you again for your engagement and recognition.

---

### Decision · Program_Chairs · 2026-04-30

**Decision:**

Accept (regular)

**Comment:**

This paper introduces PerceptionRubrics (PRB), a rubric-based evaluation framework for MLLMs that decomposes dense image perception into atomic Must-Right and Easy-Wrong criteria, addressing the saturation and compressed rankings of existing benchmarks. Reviewers initially raised concerns around methodological novelty relative to prior hallucination benchmarks, the cost and reproducibility of the pipeline, the scale of human validation, and potential bias from using the same model families in both construction and evaluation; the authors responded with detailed cost comparisons, inter-annotator agreement statistics (98.99% Must-Right confirmation rate), a role-separation argument against circularity, and ranking comparisons against VisionArena, which largely satisfied two reviewers (Keee raised score, 2wCg marked concerns fully resolved), while pVFd remained unconvinced on the novelty front. Given the broadly positive reception from three out of four reviewers and the practical value demonstrated through strong human alignment, I would like to give weak accept for this paper.